# A Multi-Region Brain Model to Elucidate the Role of Hippocampus in Spatially Embedded Decision-Making

**Yi Xie** [1 2]  **Jaedong Hwang** [1]  **Carlos Brody** [2 3]  **David Tank** [2]  **Ila Fiete** [1]

## Abstract

Brains excel at robust decision-making and data-efficient learning. Understanding the architectures and dynamics underlying these capabilities can inform inductive biases for deep learning. We present a multi-region brain model that explores the normative role of structured memory circuits in a spatially embedded binary decision-making task from neuroscience. We counterfactually compare the learning performance and neural representations of reinforcement learning (RL) agents with brain models of different interaction architectures between grid and place cells in the entorhinal cortex and hippocampus, coupled with an action-selection cortical recurrent neural network. We demonstrate that a specific architecture–where grid cells receive and jointly encode self-movement velocity signals and decision evidence increments–optimizes learning efficiency while best reproducing experimental observations relative to alternative architectures. Our findings thus suggest brain-inspired structured architectures for efficient RL. Importantly, the models make novel, testable predictions about organization and information flow within the entorhinal-hippocampal-neocortical circuit: we predict that grid cells must conjunctively encode position and evidence for effective spatial decision-making, directly motivating new neurophysiological experiments.[*]

## 1. Introduction

Deep learning has advanced through the adoption of larger datasets (Lin et al., 2014; Russakovsky et al., 2015; Schuh-

---
[*]See project page at https://minzsiure.github.io/multiregion-brain-model/. [1]Massachusetts Institute of Technology, Cambridge, MA, USA [2]Princeton Neuroscience Institute, Princeton University, Princeton, NJ, USA [3]Howard Hughes Medical Institute, USA. Correspondence to: Yi Xie <evayixie@princeton.edu>, Ila Fiete <fiete@mit.edu>.

*Proceedings of the 42nd International Conference on Machine Learning*, Vancouver, Canada. PMLR 267, 2025. Copyright 2025 by the author(s).

mann et al., 2022) and deeper architectures (Dosovitskiy et al., 2021; He et al., 2016; Jiang et al., 2023), frequently emphasizing scale over the efficiency driven by biologically inspired models (Banino et al., 2018). To bridge this gap, insights from neuroscience can inform more efficient architectures by studying how biological systems process information to make robust and adaptive decisions in dynamic and uncertain environments. The brain contains specialized circuits including the hippocampal circuit (HPC), a key set of brain areas critical for spatial, contextual, and associative learning and memory (O'Keefe, 1978; Dostrovsky & O'Keefe, 1971; Squire, 1992; Scoville & Milner, 1957). Meanwhile, cortical and subcortical regions play a central role in evidence accumulation and decision-making (Pinto et al., 2019; IBL et al., 2023).

Brain-scale neural recordings at cellular resolution, which are only recently possible, open a window into how brain regions interact with each other to perform complex tasks. Here we focus on the accumulating tower task, a widely adopted, interpretable benchmark in neuroscience for probing multi-region brain interactions underlying spatially embedded evidence accumulation and decision-making (Pinto et al., 2019; 2022; Nieh et al., 2021; Brown et al., 2024).

In the task, mice navigate an immersive virtual reality corridor, where they are stochastically presented with visual towers on both sides. At the end of the corridor, they must turn left or right depending on which side has more towers (see Figure 1D). This task requires integrating evidence: computing the difference in the total number of towers on each side ("accumulated evidence"). Task standardization enables reproducible experiments, and these in turn inform theoretical models (Lee et al., 2024; Karniol-Tambour et al., 2024) that generate testable predictions about behavior and neural dynamics. We focus on this task to build a multi-system brain model spanning memory, integration, spatial navigation, and decision circuits.

During the task, the dorsal CA1 region of the hippocampus encodes *conjunctive* cognitive maps of both the animal's location and accumulated evidence. Place fields in this region are tuned not only to spatial position but also to task-relevant accumulated evidence, meaning that individual neurons fire selectively based on both variables (Nieh et al.,

2021).

This finding is intriguing given that the task does not explicitly require spatial information for decision-making–the correct choice depends only on the relative frequency of towers on each side. It raises several key questions: Why does the hippocampus, typically associated with spatial navigation and episodic memory, represent accumulated evidence in this task? Does this suggest that the hippocampus has a broader functional role in spatially embedded decision-making tasks, even when spatial information is unnecessary for the decision? Furthermore, why are spatial and evidence representations jointly encoded in the hippocampus? These questions point to the potential involvement of the hippocampus in coordinating computations across multiple brain regions during decision-making.

Decision-making tasks are often modeled within a reinforcement learning (RL) framework (Gershman & Niv, 2015; Gershman & Daw, 2017). While these models excel in task-level performance, they often overlook structured neural architectures and dynamics observed in biological systems. For instance, deep RL approaches applied to the accumulating tower task (Mochizuki-Freeman et al., 2023; Lee et al., 2024) focus on optimizing performance post-training but fail to capture the distributed computations across brain regions and do not explain how natural systems perform efficient and robust learning.

To address these gaps, we developed a multi-region brain model that incorporates an architecturally and dynamically prestructured circuit model of the hippocampal-entorhinal system, Vector Hippocampal Scaffolded Heteroassociative Memory (Vector-HaSH) (Chandra et al., 2025). Our model extends Vector-HaSH by integrating it with cortical and subcortical regions, abstracted as a recurrent neural network (RNN) (Elman, 1990) to serve as the RL decision-making actor. This integration enables the model to function as a biologically grounded RL solver, leveraging structured memory circuits to support spatially embedded decision-making. Inspired by the vision for autonomous machine intelligence outlined in LeCun (2022), we demonstrate that integrating structured, content-addressable associative memory with neural representations is a promising approach for efficient task learning and navigation. Specifically, our work highlights the essential role of structured coding schemes, such as grid cells, in forming world models (cognitive maps) that support efficient task-solving.

We apply this framework to the accumulating tower task and test counterfactual scenarios in which the entorhinal-hippocampal networks receive different inputs. Our model generates normative predictions about tuning in grid cells, the role of entorhinal-hippocampal networks, and the conditions that give rise to efficient learning and performance of spatially embedded decision-making tasks. These findings illuminate how the brain may coordinate computations across its many substructures to flexibly and efficiently tackle complex challenges.

The contributions of this paper are four-fold:

- We propose and demonstrate a multi-region brain model framework that counterfactually tests the computational roles of entorhinal-hippocampal-neocortical interactions during spatially embedded decision-making tasks. Our framework makes novel experimentally verifiable predictions for neuroscience.

- The model enables systematic exploration of how neural computations shape cognitive capabilities, offering a tool to guide and interpret future neuroscience experiments.

- We predict that conjunctive position-evidence tuning in grid cells is essential to the emergence of experimentally observed conjunctive position-evidence hippocampal representations (Nieh et al., 2021).

- Finally, we demonstrate that conjunctive grid cell tuning and non-grid sensory inputs to the hippocampus are critical for learning spatially embedded contexts (model M5).

## 2. Related Works

### 2.1. Biological Evidence and Gaps on Entorhinal-Hippocampal-Neocortical Interactions

Hippocampal place cells (HPC), which encode spatial locations through their activity patterns (Dostrovsky & O'Keefe, 1971), form the basis of the cognitive map theory (O'Keefe, 1978; Fenton, 2015; Moser & Moser, 2016). This theory provides a foundational framework for understanding how flexible and intelligent behaviors arise from coordinated neuronal populations (Fenton, 2024). Cognitive maps not only enable flexible spatial navigation but also support memory organization and the construction of coherent narratives of personal experiences (O'Keefe, 1978; Tolman, 1948; Whittington et al., 2022; Fenton, 2024). Furthermore, the HPC's ability to encode internal cognitive variables, such as accumulated evidence and task-relevant information, highlights its broader role as a model system for studying internally generated cognition (Bostock et al., 1991; Nieh et al., 2021; Ólafsdóttir et al., 2015; Tavares et al., 2015).

Interactions between the HPC and the entorhinal cortex (EC) are well documented as crucial for navigation (McNaughton et al., 1996; O'Keefe, 1978) and declarative memory (Scoville & Milner, 1957; Squire, 1992). Within the medial entorhinal cortex (MEC), grid cells (Hafting et al., 2005) provide a spatial metric through their periodic, hexagonal firing fields (Krupic et al., 2012). These grid cells, along

with hippocampal place cells, form the building blocks of neural systems that support both physical navigation and abstract cognitive functions. Recent hypotheses propose that memory and planning mechanisms evolved from processes originally adapted for physical navigation, suggesting a shared computational framework for navigating both physical and mental spaces (Buzsáki & Moser, 2013). Moreover, hippocampal-prefrontal interactions have been shown to play a significant role in higher-order cognitive functions, including decision-making and planning (Eichenbaum, 2017; Preston & Eichenbaum, 2013). These established interactions inform the design of our multi-region model to uncover neural mechanisms underlying cognition.

Recent advances in experimental techniques, such as large-scale neural recordings (Jun et al., 2017; Steinmetz et al., 2021), have made it possible to investigate how thousands of neurons across multiple brain regions coordinate to function coherently (Bondy et al., 2024). However, despite these advances, understanding the mechanistic roles and interactions of individual brain regions in cognition remains a challenge. Our theoretical work bridges this gap by providing interpretable, mechanistic model "testbeds" that complement experimental findings and generate testable predictions to guide future investigations.

### 2.2. Models of Entorhinal-Hippocampal Interactions

Prominent models of entorhinal-hippocampal interactions include the Tolman-Eichenbaum Machine (TEM) (Whittington et al., 2020), a statistical generative model, and Vector-HaSH (Chandra et al., 2025), a biologically realistic, mechanistic model. Vector-HaSH separates fixed-point dynamics for pattern completion from content encoding, leveraging grid-cell scaffolds to prevent catastrophic forgetting and memory capacity cliffs. Unlike generative models, it provides a high-capacity, generalizable framework for spatial and non-spatial memory, making it well-suited to studying episodic and spatial representations.

In contrast to data-driven approaches, such as inferring brain-wide interactions with constrained RNNs (Perich et al., 2020) or disentangling shared and private latent variables across regions (Koukuntla et al., 2024), our mechanistic model provides interpretable hypotheses about entorhinal-hippocampal interactions. By integrating Vector-HaSH with multi-region dynamics, our approach bridges experimental findings with theoretical predictions, advancing understanding of distributed neural computations.

### 2.3. Bidirectional Insights Between Deep Learning and Neuroscience

Machine learning-based frameworks are increasingly applied in neuroscience studies (Richards et al., 2019). For instance, deep neural networks have emerged as plausible

models of the brain (Sacramento et al., 2018; Whittington & Bogacz, 2017), mimicking representational transformations in primate perceptual systems (Bashivan et al., 2019; Kell et al., 2018). These models often exhibit classic behavioral and neurophysiological phenomena when trained on tasks similar to those performed by animals (Banino et al., 2018; Pospisil et al., 2018; Wang et al., 2018). Complementarily, Yamins & DiCarlo (2016) demonstrated correlations between artificial neural network (ANN) representations and neuronal activity in the monkey visual cortex during image classification tasks, informing the design of brain-like ANNs (Kubilius et al., 2019; Zhuang et al., 2021). Such brain-inspired approaches provide value to both neuroscience and machine learning. For example, while navigation is fundamental for humans, it remains challenging for ANNs (Mirowski et al., 2017). Leveraging grid cell-like representations, critical for mammalian navigation, Banino et al. (2018) developed a deep RL agent with navigation abilities resembling those of primates.

## 3. Methods

### 3.1. Entorhinal-Hippocampal-Neocortical Spatial Decision Model

Our multi-region brain model integrates a cortical circuit, abstracted into an action-selection RNN policy, with a pre-structured entorhinal-hippocampal circuit inspired by Chandra et al. (2025), which incorporates bidirectional computations between grid cells and place cells to associate, encode, and learn environmental information. As shown in Chandra et al. (2025) and Fig 1A (purple and orange), the entorhinal-hippocampal memory scaffold features a bipartite architecture comprising hidden (hippocampal) and label (grid cell) layers. The scaffold's design is based on established and inferred recurrent connectivity patterns between the MEC and HPC (Witter & Groenewegen, 1984; Amaral & Witter, 1989; Witter & Amaral, 1991; Witter et al., 2017) and among grid cells in the MEC (Burak & Fiete, 2009).

Connections from grid cells to the hippocampus ($\mathbf{W}_{hg}$) are fixed and random, while connections from the hippocampus to grid cells ($\mathbf{W}_{gh}$) are set through associative learning and remain fixed thereafter. Connections between the HPC and non-grid lateral entorhinal cortex (LEC) ($\mathbf{W}_{hs}$ and $\mathbf{W}_{sh}$) are learned bidirectionally through associative learning. The grid cell layer operates as a $k$-hot modular vector, constrained by local recurrent inhibition, where $k$ reflects the number of one-hot grid modules. Each module has a unique periodicity, and velocity inputs (*e.g.*, position and evidence) drive the phase progression of each module within its 2D representational space (*i.e.*, a 2D torus).

We build upon the architecture proposed by Chandra et al. (2025) (Fig 1A), termed *Vector-HaSH+* (Fig 1B). Unlike the

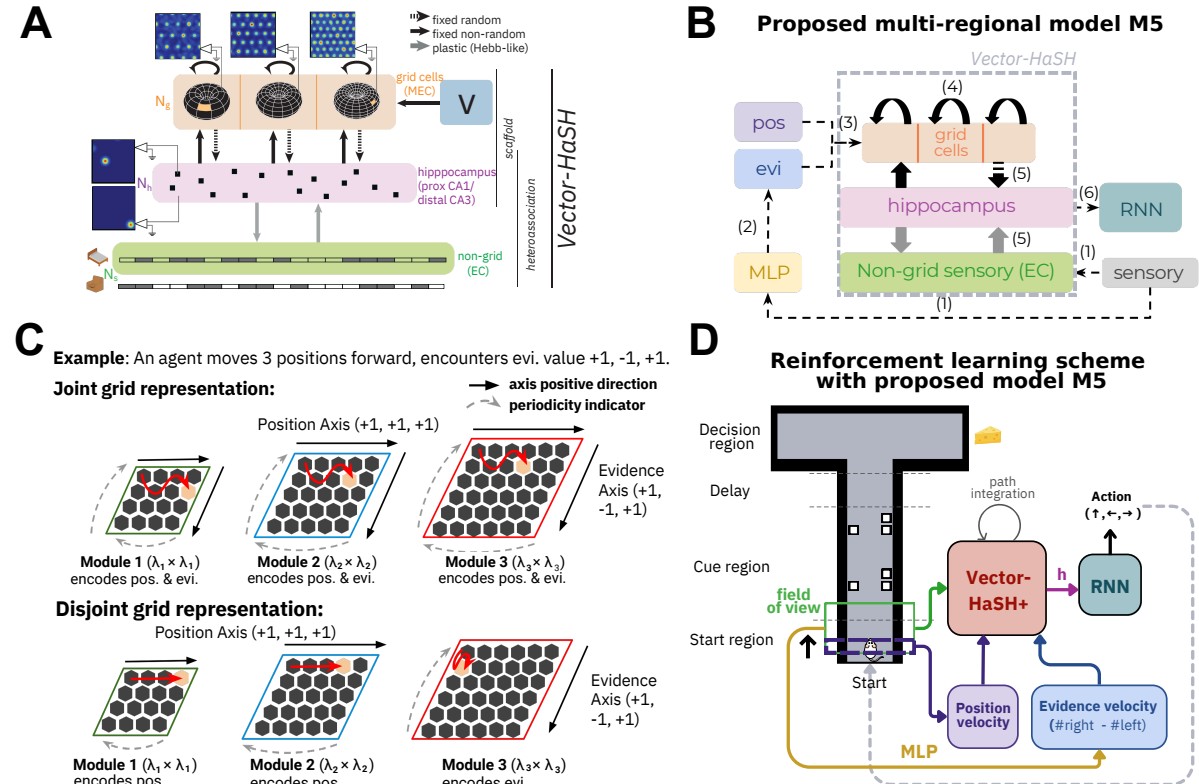

Figure 1. **Task schematics. (A)** Schematic of Vector-HaSH (Chandra et al., 2025), the basis of our model architecture. **(B)** Schematic of the Vector-HaSH+ circuit, which we propose to model and investigate the neural computation process for spatially embedded decision-making tasks. The numbers in parentheses are the order of computation. **(C)** Schematic of grid cell code, for a specific example of an agent moving 3 positions forward while encountering evidence value $+1$, $-1$, and $+1$. Here, we assume the grid state is initialized at the top left corner, but the coding scheme is invariant regardless of the initial state. A *joint* grid representation (top) utilizes both axes of the grid module 2D space for both task variables, position and accumulated evidence, yielding a wiggling activation pattern (red arrows). A *disjoint* grid representation (bottom) encodes task variables in separate modules, such that each grid module only fires along one axis (red arrows). The periodicity of each module is indicated by dashed gray arrows, as the representation space of a grid module is effectively a 2D torus. **(D)** Schematic of the RL setup in which an agent navigates a virtual T-maze with towers appearing on both sides, and a reward is given when it turns to the side with more towers in the end. The agent has some field of view ahead, and the visual sensory information is communicated to HPC through MEC and/or non-grid LEC. The HPC code is then mapped by an RNN policy (cortex) to select an action. The action updates the agent's position, which updates the sensory input and then the grid states. This process repeats until task termination.

original Vector-HaSH, *Vector-HaSH+* provides flexibility in hippocampal readouts, allowing them to receive projections from both grid cells and non-grid sensory inputs simultaneously or from just one source (Fig 1B, orange and green). Grid states are updated by task-relevant velocity inputs, either across all modules or selectively along specific axes in some modules (see Fig 1C and Appendix A.1). A multilayer perceptron (MLP) processes sensory inputs to extract evidence velocity (Fig 1B, yellow), which informs grid cell updates. Similar to Hwang et al. (2023); Wang et al. (2024), the resulting hippocampal vector is the input to the RNN policy, which is trained using RL policy gradient method to make action decisions (see Fig 1D and Appendix A.3 for a detailed step-by-step procedure).

While the MEC, HPC, and LEC all interact with the cortex

biologically (Preston & Eichenbaum, 2013; Eichenbaum, 2017; Canto et al., 2008), we model the hippocampal vector as the primary cortical input for simplicity. This simplification assumes the hippocampal vector alone is sufficient for learning the task, while allowing us to test hypotheses about conjunctive hippocampal representations observed in Nieh et al. (2021), providing a foundation for further investigations. For instance, future studies could evaluate the computational benefits of various combinations of {MEC, HPC, LEC} inputs to the cortex in enabling generalization and efficient learning.

### 3.2. Model Setup

We describe the model formally in the context of the accumulating tower task. As the agent navigates in space

at time $t$, it processes sensory information from the left and right visual fields, $\vec{f}_L$ and $\vec{f}_R$, which is projected through the dorsal visual stream to downstream processing regions. This results in a sensory vector in LEC modeled by $\vec{s}(t) = \mathbf{W}_R \cdot \vec{f}_L(t) + \mathbf{W}_L \cdot \vec{f}_R(t)$, representing a weighted integration of the two fields that can be used for further computations. We assume a simple concatenation of $\vec{f}_L(t)$ and $\vec{f}_R(t)$, but this setup provides flexibility for modeling ablation studies, *e.g.*, simulating the effects of optogenetically inhibiting one hemisphere. The downstream computation includes velocity prediction that updates grid cell states and projection into HPC.

Following Chandra et al. (2025), the MEC layer of the model contains $k$ one-hot grid cell modules, each is a binary-valued periodic function on a 2D discretized hexagonal lattice space with periodicity $\lambda$. Thus, each module state is a vector of dimension $\lambda \times \lambda$. The module states are concatenated to form a collective grid state $\vec{g} \in \{0,1\}^{N_g}$, where the vector length $N_g = \sum_M \lambda_M^2$.

The grid cell state is updated through continuous attractor recurrence dynamics (Burak & Fiete, 2009), where a module-wise winner-take-all mechanism, $CAN[\cdot]$, shifts each grid module based on position and evidence velocity signals $v(t)$ informed by $\vec{s}(t)$. We model the velocity estimation as an MLP (Rosenblatt, 1958) that processes sensory inputs for simplicity, representing a form of visual-vestibular integration (DeAngelis & Angelaki, 2012). Following Chandra et al. (2025), we assume this process occurs externally to the entorhinal-hippocampal circuit. This simplification does not affect our results and is beyond the scope of this framework.

The grid cell state update at time $t$ is thus formalized as

$$\vec{g}(t+1) = \text{CAN}\big[\vec{g}(t),\, v(t)\big]. \tag{1}$$

The full implementation of $\text{CAN}[\cdot]$ is provided in Appendix A.5.

The grid cell layer and the non-grid sensory layer project onto the HPC layer, such that the hippocampal activities are

$$\vec{h}_{\text{mix}}(t+1) = \text{ReLU}[\mathbf{W}_{hs} \cdot \vec{s}(t) + \mathbf{W}_{hg} \cdot \vec{g}(t+1)]. \tag{2}$$

We also test the variants of hippocampal coding, in which only the grid cell layer projects onto the HPC layer, such that the hippocampal activities are

$$\vec{h}_{\text{nonmix}}(t+1) = \text{ReLU}[\mathbf{W}_{hg} \cdot \vec{g}(t+1)]. \tag{3}$$

The connectivity between the HPC layer and the EC layer is updated in both cases as pseudo-inverse ($^+$) learned heteroassociative weights,

$$\mathbf{W}_{hs} = \mathbf{H}\mathbf{S}^+, \tag{4}$$

$$\mathbf{W}_{sh} = \mathbf{S}\mathbf{H}^+, \tag{5}$$

*Table 1.* Overview of how model variants correspond to alternative hypotheses of neural coding and information flow based on evidence source. Our final model, M5, is marked with $*$.

| Models of hypotheses (Source of Evidence) | Not grid cells | grid cells |
|---|---|---|
| Not sensory | M1 | M3 |
| sensory | M2 | M4, M5* |

*Table 2.* Summary of the neural coding and information flow in each model variant. Our final model, M5, is marked with $*$.

| Model | Grid cell code | Place cell code | MLP input | RNN input |
|---|---|---|---|---|
| M0 | - | - | - | s |
| M0+ | - | - | s | s & $v_{pos}$ & $v_{evi}$ |
| M1 | pos. | g | - | p |
| M2 | pos. | g & s | - | p |
| M3 | joint pos. & evi. | g | s | p |
| M4 | disjoint pos. & evi. | g & s | s | p |
| M5* | joint pos. & evi. | g & s | s | p |

where $\mathbf{H}$ is a $N_h \times N_{patts}$ matrix with $N_{patts}$ hippocampal states, each of length $N_h$, and $\mathbf{S}$ is a $N_s \times N_{patts}$ matrix with columns as the encoded sensory inputs of length $N_s$.

We modeled variants with recurrence in the HPC layer (namely, the CA3 recurrence (Sammons et al., 2024)) in Appendix F, which does not alter the conclusions in the main paper. The hippocampal state $\vec{h}$ in Eqn 2 (or Eqn 3) is the readout of the entorhinal-hippocampal circuit to the cortex (under the modeling rationale explained in Section 3.1), which is an action-selection RNN policy trained through policy gradient under reinforcement learning. Please refer to Appendix A.3 for what one step of the agent in the environment entails among the involved brain regions.

## 4. Alternative Multi-Region Interaction Hypotheses

Nieh et al. (2021) observed conjunctive coding of both accumulated evidence (cognitive variable) and position (physical variable) in the hippocampus when mice perform the accumulating tower task, suggesting that the hippocampus also performs a general computation, rather than merely responding to features of external stimulus such as space (O'Keefe & Burgess, 1996). This discovery highlights the need for mechanistic modeling at an appropriate scale to explore how interactions across multiple brain regions contribute to internally generated cognition. Such processes enable individuals to flexibly navigate spaces and organize, relate, and integrate experiences, objects, and events.

We use the accumulating tower task as a minimal framework to hypothesize three potential mechanisms underlying the conjunctive HPC code of physical and cognitive variables:

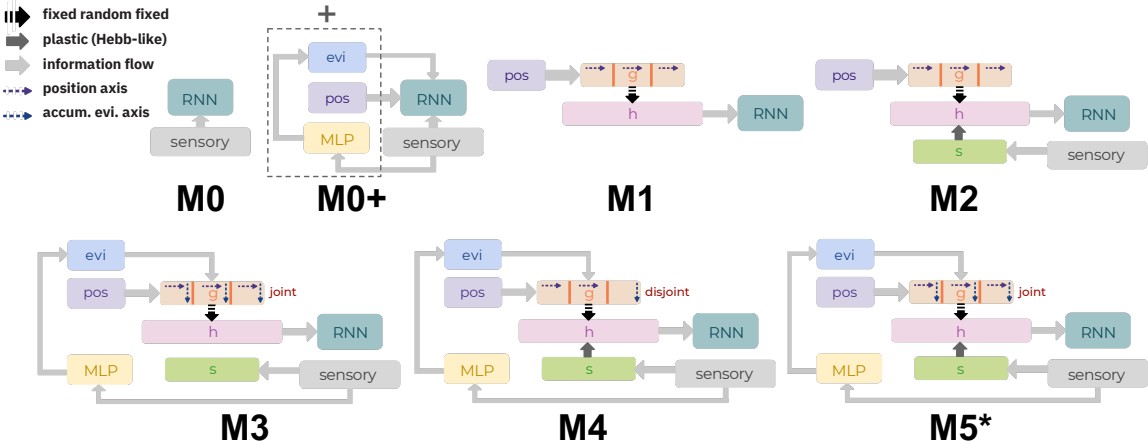

*Figure 2.* **Model schematics.** Counterfactual models of hypotheses on neural code and information flow, as detailed in Tables 1 and 2.

- Grid cells encode position, aligning with the prevailing view (Moser et al., 2008), and the conjunctive encoding in the HPC arises from sensory input provided by non-grid LEC neurons (M2).

- Grid cells co-tune to both position and evidence, a phenomenon that has not been extensively investigated experimentally to the best of our knowledge. The LEC pathway is neither necessary nor relevant (M3).

- Grid cells co-tune to both position and evidence, and the EC pathway contributes to the formation of the conjunctive hippocampal code (M4, M5).

Notably, there are two possible mechanisms for grid cell tuning of accumulated evidence and position:

- **Joint Integration Model:** Grid cell modules each encode a combination of evidence and position by leveraging their 2D toroidal attractor network. This implies the simultaneous representation of spatial and cognitive variables within the same grid modules (M3 & M5, see Fig 1C top).

- **Disjoint Integration Model:** Individual grid cell modules each encode distinct task variables. Specifically, some grid cell modules exclusively encode position, while others exclusively encode evidence (M4, see Fig 1C bottom).

Our framework provides a systematic approach to evaluate the proposed hypotheses shown in Tables 1 and 2, and Fig 2. This evaluation includes (a) quantitative analyses of learning performance and behavioral outcomes (Section 5.1) and (b) qualitative alignment with experimental findings (Section 5.2). Furthermore, we hypothesize the roles of individual brain regions in spatially embedded decision-making tasks by analyzing neural representations (Section 5.3).

## 5. Results

### 5.1. Joint Integration Model Induces Efficient Learning

We compare the performance of different model variants in terms of cumulative success rate and exploration efficiency during training (Fig 3). Additionally, as shown in Appendices G and H, our findings remain consistent after tuning the learning rate or matching the number of parameters in M0 and M0+ to that of M5. Our results show that agents fail to solve the task when grid cells do not encode evidence (Fig 3A, orange and green), highlighting the importance of MEC in integrating cognitive variables. Moreover, RNN-only baselines (M0 and M0+, in blue and black, respectively), with the same number of neurons as M1–M5, perform poorly even when supplied with positional and velocity information (M0+, in black), suggesting that the EC-HPC network is critical for temporal integration. Additionally, models with jointly tuned grid cells (red and brown) learn more efficiently than those with disjointly tuned grid cells (purple). This phenomenon lacks an immediately clear computational explanation, which we investigate in Section 5.2.

Interestingly, when sensory information is projected to the hippocampus (M4, M5), the learning performance becomes more variable relative to M3 (Fig 3A), possibly because mixing place codes with sensory signals complicates the representation. However, including sensory information in HPC increases exploration efficiency (brown in Fig 3B), presumably because it captures the nuances in the environment—such as wall positions—complement the rigid information encoded by grid cells. In support of this view, M3, a variant of M5 without sensory projections, requires longer navigation times (red in Fig 3B). Please refer to Appendix A.1 for implementation details.

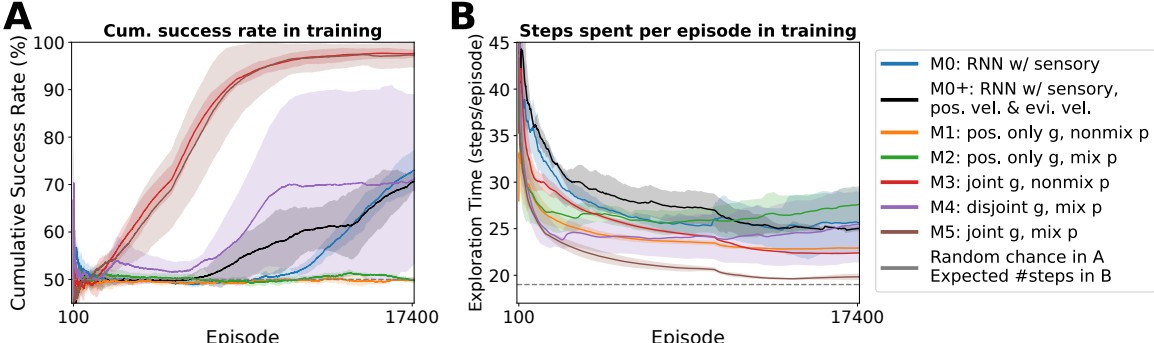

*Figure 3.* **Learning performance measured by cumulative success rate and exploration efficiency over the course of training for all model variants.** We present the mean and standard deviation of these metrics across three trials. Baselines are indicated by dashed gray lines. In (A), using a window size of $5,000$ episodes, we observe efficient learning in models with jointly tuned grid cells (M3, in red; M5, in brown). In (B), with a window size of $10,000$ episodes, M5 demonstrates its ability to effectively leverage spatial information, navigating the maze more quickly (brown). For clarity, data from the first 100 episodes are excluded due to initial instability.

## 5.2. Joint Integration Model Predicts Evidence-Position Co-Tuning in Grid Cells

As shown among simulated hypotheses, joint tuning of position and evidence in the MEC promotes efficient learning, while the sensory pathway enhances efficient navigation. Here, we further analyze the relationship between grid cell computations and place cell firing patterns. Our results reveal that model variants capable of efficient task learning exhibit firing fields closely resembling the experimental observations in Nieh et al. (2021). This alignment makes a clear prediction that the conjunctive grid cell representations give rise to the joint encoding of spatial and cognitive variables in the HPC.

### 5.2.1. JOINT INTEGRATION MODEL EXHIBITS EXPERIMENTALLY ALIGNED HPC FIELDS

Nieh et al. (2021) demonstrated experimentally that individual CA1 neurons encode both position and accumulated evidence. This interdependence implies that trials leading to the same final decision would evoke distinct hippocampal firing sequences, as the agent traverses different tower/evidence configurations (Fig 4, left). Consequently, smaller firing fields would partition the evidence dimension within the Evidence (E) × Position (Y) space of hippocampal activity.

We found that joint integration models (M5 in Fig 4, bottom right; M3 in Appendix C) successfully replicate the $E \times Y$ place fields observed by Nieh et al. (2021). In contrast, models lacking joint integration, such as M4 (Fig 4, top right), fail to reproduce this behavior, instead exhibiting stripe-like firing patterns indicative of independent representations of position and evidence. In Appendix I, we additionally demonstrate that smoothing hippocampal activity reveals more localized and stereotyped tuning, consistent with experimental observations.

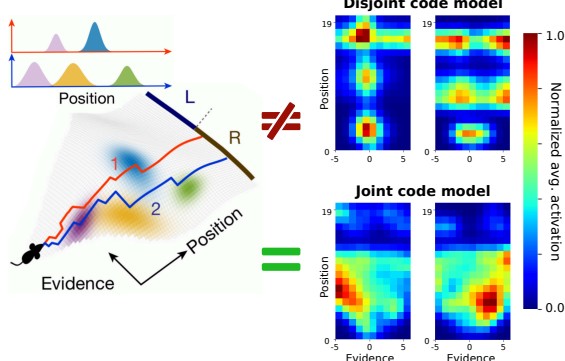

*Figure 4.* **Place cell tuning during the task.** The schematic plot (left, adapted from Nieh et al. (2021)), and the firing fields of selective hippocampal neurons in models M4 (right, top) and M5 (right, bottom). The firing field of each hippocampal cell is determined by averaging smoothed neural activity across trials and normalizing within the cell. See Appendix C for raw firing fields without smoothing and Appendix I for smoothing details using $\sigma_1, \sigma_2 = 1$. In Nieh et al. (2021), since hippocampal cells have a conjunctive code of evidence (E) and position (Y), smaller firing fields effectively partition the evidence dimension in the E × Y space. Notably, only models with jointly tuned grid cells exhibit conjunctive place fields in the E × Y space (right, bottom).

### 5.2.2. JOINT INTEGRATION MODEL EXHIBITS BOTH CHOICE-SPECIFIC FIELDS & EVIDENCE FIELDS

Here, we demonstrate that only the joint integration model aligns perfectly with the experimental findings, strongly supporting its role in governing the neural computations underlying place cell behaviors described in Nieh et al. (2021). In this model, grid cells jointly encode evidence and position, enabling the HPC to create integrated maps.

**Only joint integration models exhibit choice-specific neurons** Nieh et al. (2021) observed that CA1 neurons exhibit choice-specific place cell sequences when sorted by their

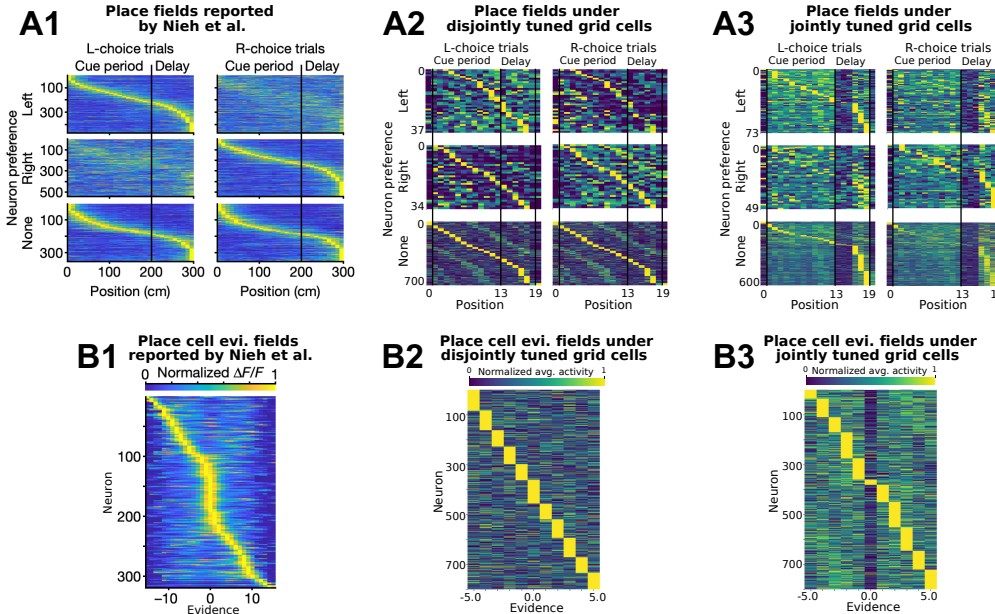

*Figure 5.* **Choice-specific place cell sequences & evidence fields.** Only models with conjunctive grid code exhibit choice-specific place cell sequences observed in Nieh et al. (2021). Activation is averaged and normalized in each cell. We compare results from (1) Nieh et al. (2021), (2) disjoint integration model M4, and (3) joint integration model M5, respectively. **(A)** Choice-specific place cell sequences. Cells are categorized into left-choice-preferring (top), right-choice-preferring (middle), and non-preferring (bottom) based on the significance of mutual information; within each row, cells are sorted by peak activities of the respective neurons of preferred choices. **(B)** Firing fields of place cells in accumulated evidence space, sorted by the positions of peak activities.

peak activity positions. To ensure a fair comparison, we analyze our models using the same approach, computing the mutual information (see Appendix B) between each cell's activity and the agent's position during left- and right-choice trials, and comparing these results to a shuffled dataset for their significance. Our analysis illustrates that only joint integration models (M3, M5) have a subset of place cells that are choice-specific under this metric (Fig 5, A3; Fig 9, B), closely matching experimental observations (Fig 5, A1). In contrast, the disjoint integration model does not exhibit choice-specific place cell sequences (Fig 5, A2).

**Place cells form firing fields in evidence space**   Similarly, we measure the mutual information between accumulated evidence and the neural activity of each place cell. As expected, when grid cells encode evidence, place cells then form firing fields in evidence space, spanning small segments of evidence values, consistent with Fig 4, left. Conversely, place cells fail to form evidence fields in models M1 and M2, where evidence information is either absent or originates from the LEC instead of MEC (see Appendix D).

## 5.3. Only Joint Integration Model With Activated EC Pathway Exhibits Well-Separated Low-Dimensional Co-representation of Task Variables

We performed Principal Component Analysis (PCA) on hippocampal and cortical activities to assess whether task variables form visually separable clusters in low-dimensional principal component (PC) space. The presence of such low-dimensional representations would provide insight into the functional roles of specific brain regions and the computational strategies employed by different model variants.

We showed that only the joint integration model with an activated LEC pathway (M5) exhibits distinct, visually separable clusters of hippocampal activity in PC space for both position (Fig 6, B1) and local evidence velocity (#R-#L towers at a position, Fig 6, B2). This contrasts sharply with other model variants (Figs 6, A1, A2, and Appendix E). Interestingly, we did not observe separability in accumulated evidence within the first three PCs of hippocampal activity. This is counterintuitive, given that grid cells encode and communicate accumulated evidence to the HPC. Since hippocampal neurons are projection neurons (Fox & Ranck Jr, 1975; 1981), the source and encoding mechanism of local evidence velocity in the HPC warrant further investigation. Future studies could analyze existing experimental data to validate the predicted hippocampal role in representing local evidence velocity and conduct ablation studies using the proposed model to understand the underlying mechanisms. Together, these findings underscore the importance of sensory inputs from the LEC in generating cohesive, low-dimensional representations of task variables in the hippocampus, which are critical for efficient learning and spatial navigation (Fig 3).

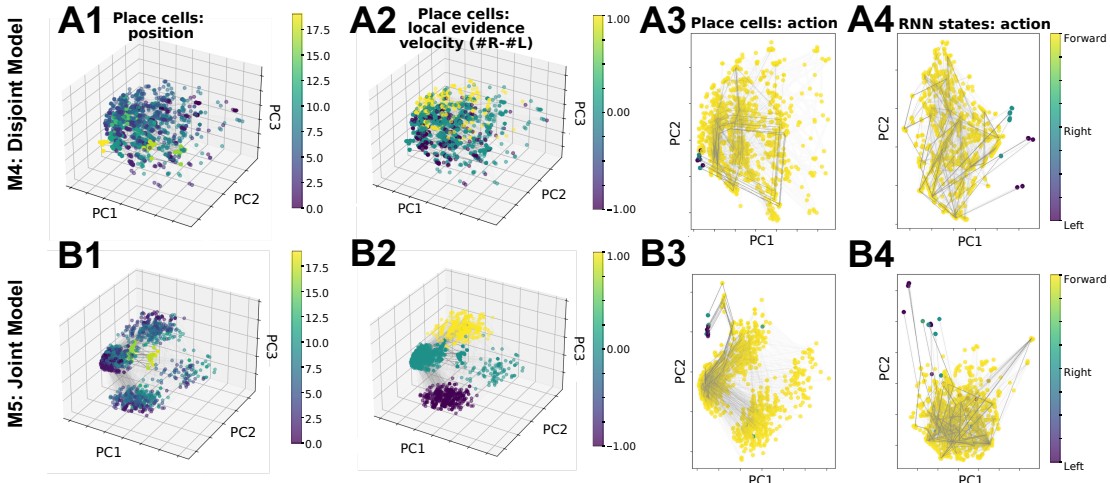

*Figure 6.* **Separability of hippocampal (column 1, 2) and cortical representations (column 3, 4) in low-dimensional PC space in M4 (row A) and M5 (row B).** Joint grid cell code with activated EC-HPC pathway (M5) uniquely leads to separable, low-dimensional hippocampal representations when colored by position (B1) and local evidence velocity (#R-#L per position, B2). We did not observe other separable task variables in the first 3 PCs of hippocampal activities in all variants (Appendix E). We observe the separability of actions in hippocampal and cortical activities in PC space in both M4 and M5 (columns 3 and 4). The gray lines indicate the trace of temporal trajectories.

## 6. Discussion

Our work predicts that grid cells jointly encode spatial and task-relevant information, and that this conjunctive coding, along with LEC sensory input to the hippocampus, facilitates efficient decision-making in spatial contexts. Moreover, our findings indicate that conjunctive grid coding is essential for replicating experimental results observed in Nieh et al. (2021), offering new insight into the prevailing view that grid cells primarily support spatial representations (Moser et al., 2008).

We derive this prediction by proposing and testing counterfactual neural codes and information flow in variants of our multi-region brain models, which integrate a prestructured entorhinal-hippocampal circuit (Chandra et al., 2025) with a cortical action-selecting recurrent neural network (RNN). These models are evaluated based on their task performance and hippocampal representations in the accumulating tower task (Nieh et al., 2021), formulated as a RL problem.

While the CA3 region of the hippocampus is known to exhibit recurrent dynamics (Sammons et al., 2024), we follow Chandra et al. (2025) in assuming, based on anatomical evidence (Donato et al., 2017), that structured, input-driven coding in the MEC plays a primary role, as it matures first and drives the development of hippocampal circuits. In Appendix F, we test variants of models M2 and M4 incorporating CA3 recurrence and find that this modification neither reproduces the experimentally observed conjunctive hippocampal code in Nieh et al. (2021) nor alters the conclusions drawn in the main paper. Building on these results, future studies could extend our framework by systematically

ablating models M1–M5 to test the computational roles of CA3 recurrence and other mechanisms. Concurrently, we are collecting neurophysiological data to directly test our falsifiable predictions and assess whether CA3 recurrence further refines hippocampal place cell tuning.

Taken together, our findings demonstrate that conjunctive grid coding is fundamental to spatially embedded decision-making, supporting the hypothesis (Buzsáki & Moser, 2013) that spatial and cognitive processes are deeply interconnected within the brain's navigation and memory systems. More broadly, neural algorithms that support path integration and spatial navigation may be repurposed for abstract cognitive functions, suggesting that the hippocampal-entorhinal network facilitates both physical navigation and decision-making based on internal cognitive states.

In conclusion, we presented a comprehensive testbed for exploring how the hippocampal-entorhinal-neocortical network integrates physical and cognitive information to build flexible neural representations that facilitate learning, decision-making, and navigation. This framework provides a foundation for future wet lab studies to test clear, falsifiable predictions while minimizing reliance on invasive and resource-intensive animal experiments, offering a platform to investigate the links between physical navigation and abstract cognitive processes. Furthermore, it highlights the mutual benefits of integrating machine learning and neuroscience in advancing our understanding of neural phenomena and guiding future research. This synergy underscores the transformative potential of neuro-inspired artificial intelligence.

## Acknowledgments

Ila Fiete is supported by the Office of Naval Research, the Howard Hughes Medical Institute (HHMI), and NIH (NIMHMH129046). Carlos Brody and David Tank are supported by NIH (U19NS132720). We are grateful to Dr. Sarthak Chandra, Dr. Manuel Schottdorf, and the BRAIN CoGS community for their many insightful discussions. We appreciate the constructive feedback received during our presentations of preliminary results at the 2024 Conference on Cognitive Computational Neuroscience and the 2025 Society for Neuroscience annual meeting.

## Impact Statement

This work advances both machine learning and neuroscience by presenting a biologically plausible multi-region brain model as a computational framework for studying decision-making in spatial contexts in mammals. The model is a testbed for hypothesis generation, offering an efficient way to explore neural mechanisms, such as the role of conjunctive coding in hippocampal-entorhinal circuits, that are difficult to measure experimentally. By reducing the need for invasive or resource-intensive experiments, our approach has the potential to accelerate neuroscience discovery while minimizing unnecessary animal studies. Additionally, the structured neural architectures studied in this work may inform the design of more robust and interpretable machine learning systems that leverage biological principles for real-world decision-making tasks.

This work contributes to advancing the fundamental understanding of the brain and bridging the gap between neuroscience and machine learning. We do not foresee significant negative societal impacts from this research.

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

# Appendix

## A. Experimental Details

### A.1. Model

For models M1–M5, we use three grid modules with periodicities of $7, 8$, and $11$, esulting in a grid cell layer dimension of $N_g = 234 \, (= 7^2 + 8^2 + 11^2)$. We simulate 800 hippocampal cells. Both the MLP and RNN models have a learning rate of $0.0005$ and a hidden size of 32. The RNN consists of leaky units with $\alpha = 0.025$.

The grid coding scheme is hand-designed to test the counterfactual of joint versus disjoint coding, as illustrated in Fig. 1C. In general, velocity inputs update the phases of each grid module via path integration, following the Vector-HaSH implementation from Chandra et al. (2025). Evidence velocity for the grid cell modules is computed as the difference between the number of towers on the right and the number on the left at the current position. This is predicted through an MLP based on the current field of view (sensory inputs). Positional velocity, in contrast, is represented as either $0$ (stationary) or $+1$ (forward movement), without an MLP, as backward movement is not task-relevant (Nieh et al. (2021), behavioral training). While an MLP could be used for positional velocity, we opted for a simplified approach since this modification does not affect the results.

For the two standalone RNN baselines (M0 and M0+), we scale up the hidden size to $32 + N_g + N_p + N_s = 1076$ to match the total number of neurons used in M1–M5. The input to these RNN baselines consists of sensory information. The M0+ variant additionally incorporates positional velocity (whether the agent has moved) and evidence velocity (predicted by an MLP with the same setup as in M1–M5). We use a learning rate of $0.0001$ with gradient clipping to a maximum norm of 1. This adjustment was necessary because the large standalone RNNs failed to train with the $0.0005$ learning rate used in M1–M5 due to learning instabilities (*e.g.,* exploding or vanishing gradients). To address this, we conducted a hyperparameter search and selected the settings that produced the best performance.

### A.2. Environment

The accumulating tower task consists of an agent navigating a T-maze with towers positioned on both sides (Fig. 1C). The agent must decide which direction to turn at the end of the corridor, aiming to turn toward the side with more towers to receive a reward. The maze is divided into distinct regions: a start region (9% of the total length) with no towers, a cue region (61%) containing towers, and a delay and decision region (the remaining portion) without towers. This structure aligns approximately with the division used in Nieh et al. (2021). Each episode presents a unique configuration of towers, requiring the agent to traverse the corridor step by step before reaching the T-junction and making a decision. The left and right sides of the maze are encoded as vectors, where a value of 1 represents a tower, 0 indicates an empty position, and $-1$ denotes areas outside the maze. The agent has a limited field of view that allows it to perceive a certain number of positions ahead.

In each episode, the rewarded side (i.e., the side with more towers) is chosen uniformly at random, with the number of towers on that side, $x_{\text{reward}}$, is sampled from $\text{Uniform}(1, K)$, where $K$ is the maximum allowable number of towers. The non-rewarded side contains strictly fewer towers, with its count $x_{\text{non-reward}}$ drawn from $\text{Uniform}(0, x_{\text{reward}})$. At each step, the agent can take one of three possible actions: left (0), right (1), or forward (2). Before reaching the T-junction, the agent receives a small reward of $0.01$ for moving forward and a penalty of $-0.001$ for any other action. Once at the end of the maze, it receives a reward of 10 for making the correct turn, no reward for choosing the wrong direction, and a penalty of $-1$ for attempting to move forward and colliding with the wall. The episode ends when the agent turns or reaches the maximum number of decision attempts, the latter incurring an additional penalty of $-5$. For this study, we set the maze sequence length to 20, dividing the start, cue, delay, and decision regions into segments of length $\{1, 12, 6, 1\}$, respectively. The agent has a field of view spanning five positions. Training is conducted using the REINFORCE algorithm (policy gradient) (Sutton et al., 1999) until convergence.

### A.3. Single step in the task

Each step in the accumulating tower task follows a structured sequence (Fig. 1C). First, the agent perceives sensory information from its field of view. This information is processed by an MLP, which extracts an estimate of evidence velocity. The evidence velocity, along with position velocity, is then used to update the grid cell state through path integration. The updated grid representation is projected onto the hippocampal layer, alongside the non-grid sensory input, forming the

agent's internal state representation. The hippocampal code is then passed to the cortical RNN policy, which selects an action—moving left, right, or forward. Once an action is executed, the agent's position is updated accordingly. This new position brings in fresh sensory input, which is implicitly processed by the grid cell subnetwork to update the grid state. The updated grid state, along with the newly perceived sensory information, is then used to refine the hippocampal representation. This iterative process continues, with the agent repeatedly updating its internal representations and selecting actions, until it reaches the T-junction and makes a final turning decision. The agent's behavior is governed by the reward scheme outlined in Appendix A.2.

### A.4. Further details on the biological grounding of Vector-HaSH

While the relevant biological groundings of Vector-HaSH are inherited from and addressed in Chandra et al. (2025), we include a summary below on how the model enforces sparsity and spatial selectivity in place cells for completeness.

**Sparsity** As described in the main text, the projection matrix $\mathbf{W}_{hg}$ from grid cells to hippocampal (HPC) units is drawn from a standard Gaussian distribution, consistent with classical random projection models (see Methods, Chandra et al. (2025)). Due to the symmetry of the distribution, each entry has zero mean, and hence half the activations are expected to be subthreshold. Moreover, each grid module encodes its position using a one-hot representation, enforcing input sparsity via inductive bias. Nonlinear gating through ReLU is applied to $h$ (Eqns. 2, 3), further ensuring that only a sparse subset of HPC units are active for a given input. Importantly, the number of unique grid states $\left(\prod_i \lambda_i^2\right)$ is much smaller than the total number of possible HPC activation patterns $(2^{N_h})$. As a result, only a very small subset of HPC units are active for any given grid code, which enforces a highly sparse representation in the hippocampus.

**Selectivity** To ensure spatial selectivity, each sensory state is associated with a specific grid state. This is implemented by updating the weights $\mathbf{W}_{hs}$ and $\mathbf{W}_{sh}$ during training, such that the sensory input strongly modulates only a small subset of HPC units. Consequently, each sensory state drives a selective hippocampal code via its associated grid representation, thereby grounding place field formation in both sensory and grid input pathways.

### A.5. Continuous-attractor (CAN) update rule in Vector-HaSH

For completeness, here we make explicit the equations that govern the CAN() update used in Eqn. 1 (Burak & Fiete, 2009; Chandra et al., 2025) of the main text. Let $\mathbf{g}(t) \in \{0, 1\}^{N_g}$ denote the concatenated grid-code vector at time $t$ and $v(t) \in \{-1, +1\}$ the 1-D velocity signal. The CAN step is a velocity-dependent cyclic shift of each grid module, implemented by a block-diagonal *shift matrix* $\mathbf{M}(v(t))$:

$$\vec{g}(t+1) = \text{CAN}[\vec{g}(t),\, v(t)] = \mathbf{M}(v(t))\, \vec{g}(t). \tag{6}$$

**Example with two modules ($\lambda_1 = 3$, $\lambda_2 = 4$)** For illustration, consider two one-dimensional grid modules of periods $\lambda_1 = 3$ and $\lambda_2 = 4$ ($N_g = 7$). A rightward velocity $v = +1$ is realized by the block-diagonal matrix $\mathbf{U}$:

$$\mathbf{M}(+1) = \mathbf{U} = \left[\begin{array}{c|c} \mathbf{S}_3 & \mathbf{0} \\ \hline \mathbf{0} & \mathbf{S}_4 \end{array}\right], \quad \text{where} \quad \mathbf{S}_3 = \begin{bmatrix} 0 & 1 & 0 \\ 0 & 0 & 1 \\ 1 & 0 & 0 \end{bmatrix}, \mathbf{S}_4 = \begin{bmatrix} 0 & 1 & 0 & 0 \\ 0 & 0 & 1 & 0 \\ 0 & 0 & 0 & 1 \\ 1 & 0 & 0 & 0 \end{bmatrix}.$$

A leftward step ($v = -1$) is obtained with $\mathbf{M}(-1) = \mathbf{U}^\mathsf{T}$.

**Formal definition** Let $\lambda_k$ be the periodicity of module $k$ and $X_k = \sum_{l=1}^{k} \lambda_l$ (with $X_0 = 0$) the cumulative offset. For indices $i, j$ belonging to the same module ($X_{k-1} \leq i, j < X_k$) we set

$$M_{ij}(v) = \begin{cases} 1, & \text{if } (j - X_{k-1}) \bmod \lambda_k = (i + v) \bmod \lambda_k, \\ 0, & \text{otherwise.} \end{cases} \tag{7}$$

All cross-module blocks are zero, making $\mathbf{M}(v)$ block-diagonal.

**Extension to 2-D** For two-dimensional Vector-HaSH, we apply the same 1-D rule independently to the $x$- and $y$-components and construct the full 2-D shift via a Kronecker product of the corresponding 1-D shift matrices.

# B. Mutual information

## B.1. Mutual information analysis

We follow the mutual information analysis in Nieh et al. (2021). Here we reiterate this procedure for completeness. For each neuron, we evaluate the mutual information metric defined in Skaggs et al. (1992),

$$I = \int_x \lambda(x) \log_2 \frac{\lambda(x)}{\lambda} p(x) dx,$$

in which $I$ is the mutual information rate of the neuron in bits per section, $x$ is the spatial location (or accumulated evidence) of the agent, $\lambda(x)$ is the mean firing rate of the neuron at location (accumulated evidence) $x$, $p(x)$ is the probability density of the agent occupying location (accumulated evidence) $x$ and $\lambda = \int_x \lambda(x) p(x) dx$ is the overall mean firing activity of the neuron.

## B.2. Scatterplots of mutual information

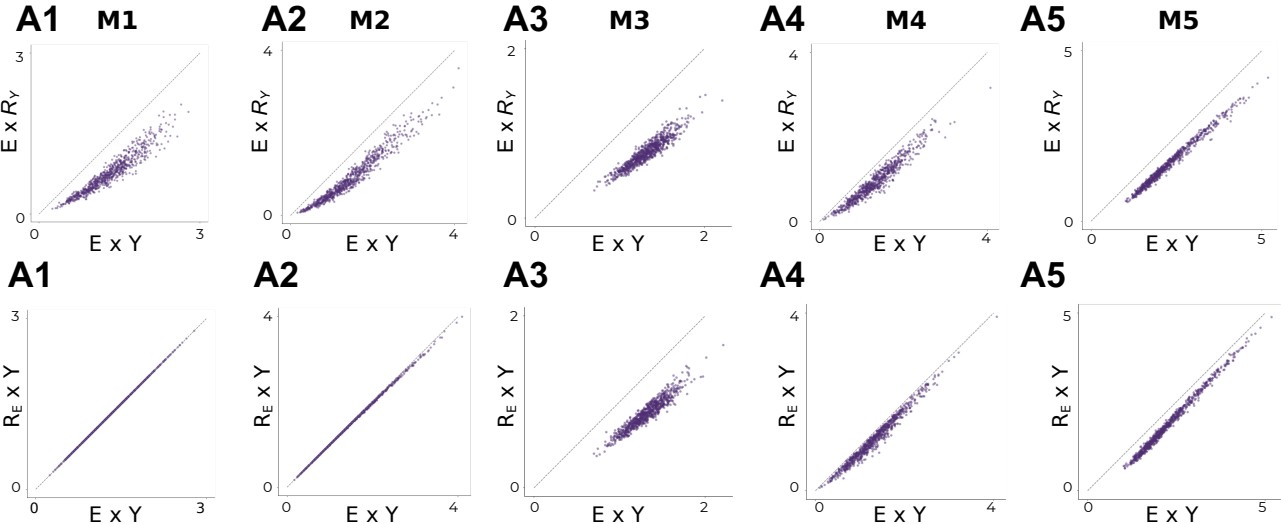

*Figure* 7. Scatterplots of the hippocampal mutual information in $E \times R_Y$ space versus $E \times Y$ space (top row), and scatterplots of mutual information in $R_E \times Y$ space versus $E \times Y$ space (bottom row). We show data for all model variants M1 to M5, in the order of panel A to $E$ respectively. We observe evidence and position interact to provide meaningful information in M3, M4, and M5 (when grid cells co-tune position and evidence), while M1 and M2 rely on information of position (when grid cells tune evidence only, and there is either no or some sensory information projected into the hippocampus). Here, $R_Y$ is a randomized position, generated by randomly sampling from the $Y$ distribution that corresponds to the non-randomized E value of the cell. A similar procedure is performed for generating the $R_E \times Y$ variables. More details of the procedure are described in the Mutual Information Analysis section of Nieh et al. (2021).

## C. Hippocampal firing fields within $E \times Y$ space in model variants

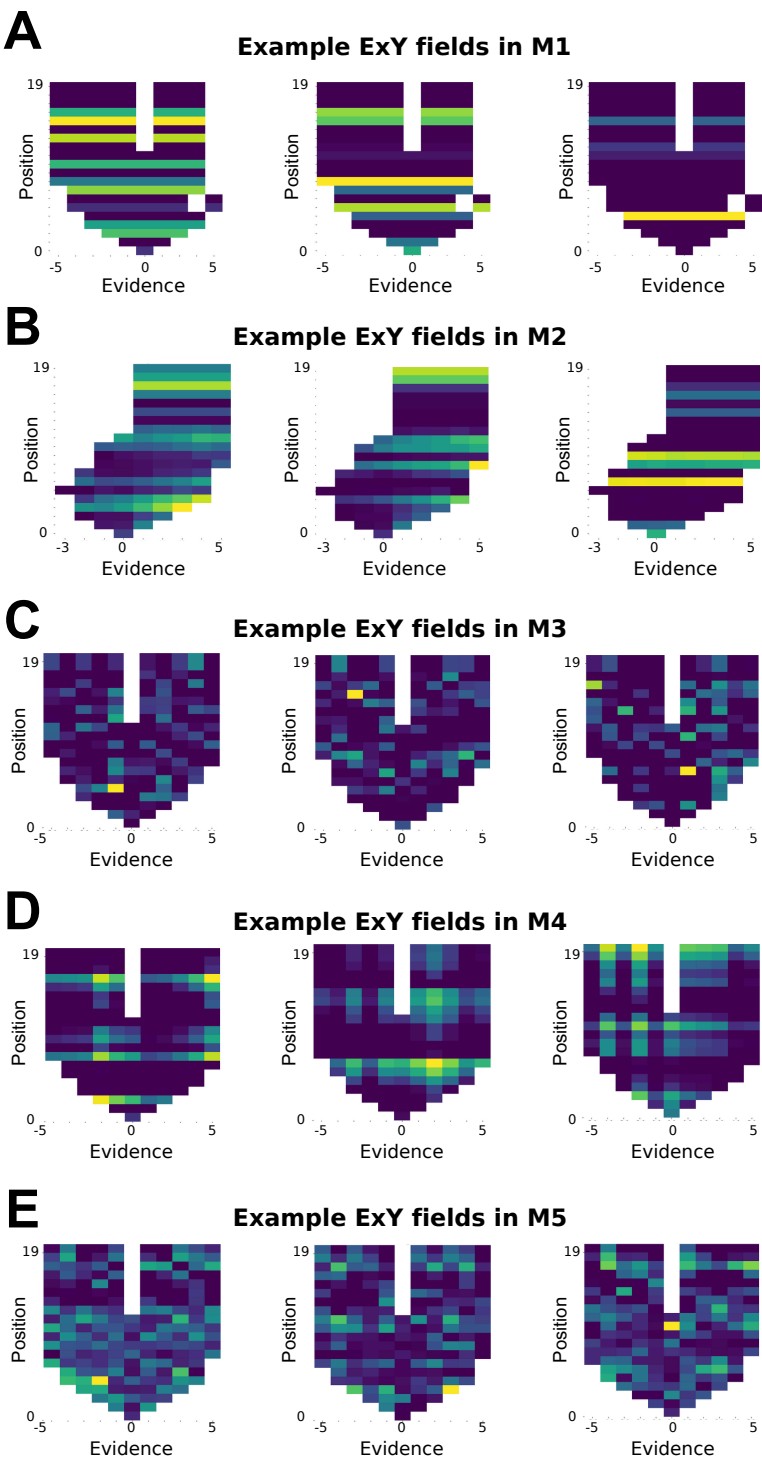

*Figure 8.* Example hippocampal firing fields in $E \times Y$ space in M1 (A), M2 (B), M3 (C), M4 (D), and M5 (E). We observe that the firing fields of M1 and M2 (A, B) do not depend on evidence with stripe patterns. M2 firing fields occasionally have some amount of gradient, a potential artifact of sensory injection, similar to the firing fields of M4 and M5 (D, E). M3 (C) firing fields exhibit conjoint tuning of position and evidence and have no apparent gradient artifacts.

## D. Hippocampal evidence and place fields in model variants

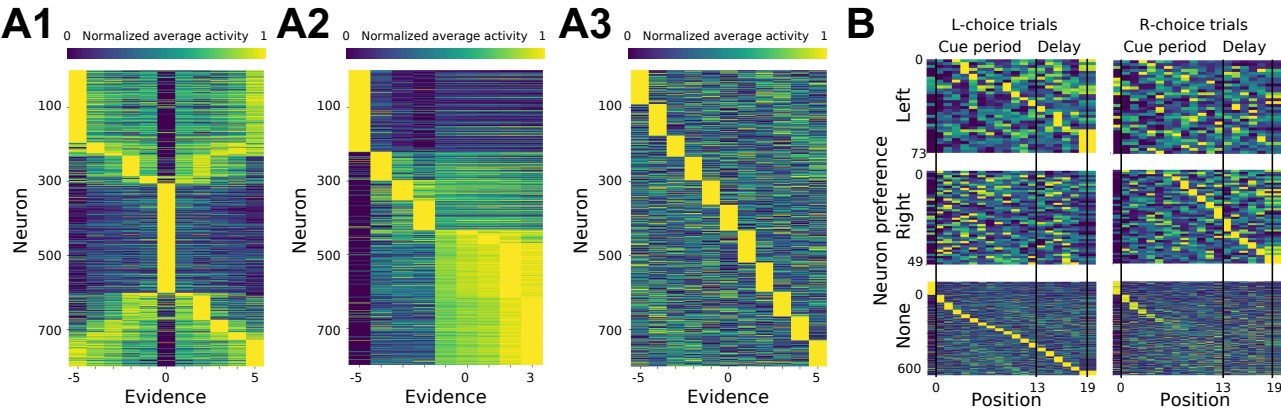

*Figure 9.* Hippocampal firing fields in evidence (A) and in space (B), for M1 (A1), M2 (A2), and M3 (A3, B). We see M1 and M2 do not have firing fields in evidence (A1, A2), while M3 does (A3). Furthermore, M3 contains choice-specific place fields (B) similar to M4 (Fig 5, A3), implying that joint tuning of position and evidence in grid cells is key to forming a conjoint hippocampal map.

## E. HPC and RNN PC representation in model variants

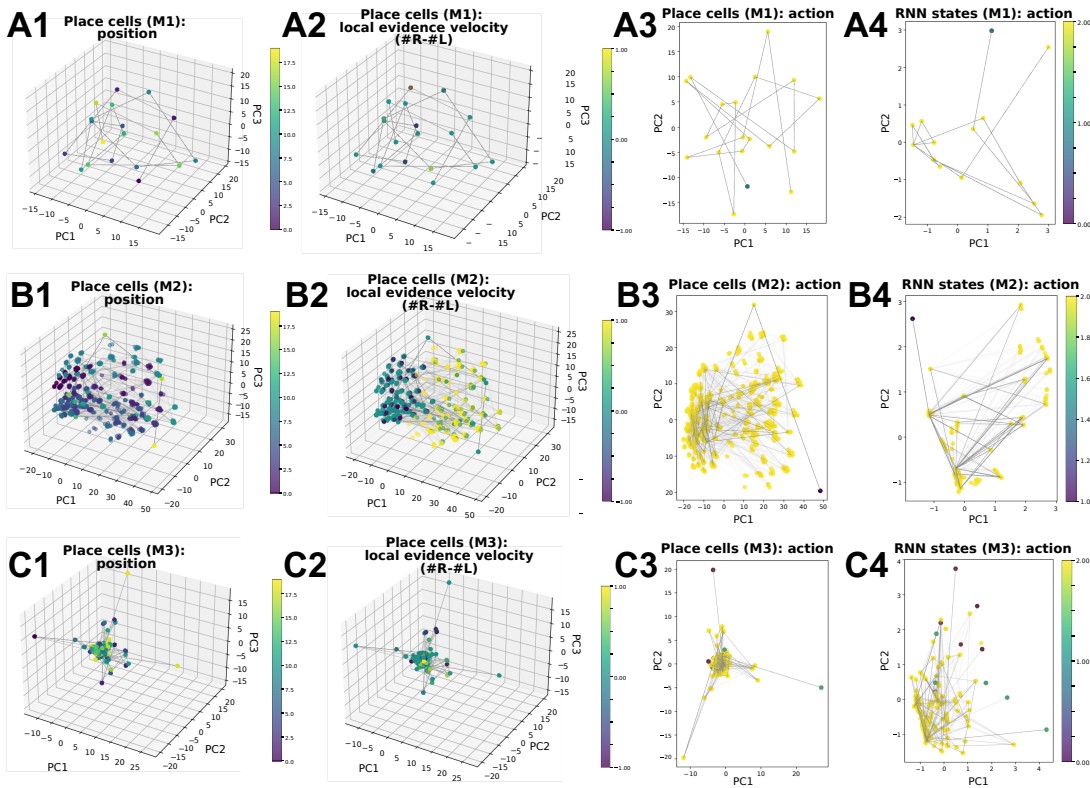

*Figure 10.* Low-dimensional representation of hippocampal and RNN activities in PC space, shown for M1 (row A), M2 (row B), M3 (row C). We show the representations colored according to selective task variables, specifically position, local evidence velocity, and action, in which M4 shows clear separation (in correspondence to Fig 6). Other variables visualized in HPC and RNN activity PC space include accumulated evidence, position changes, left-/right-choice trials, total evidence of the trial, and ground truth action; we observe no visual separation.

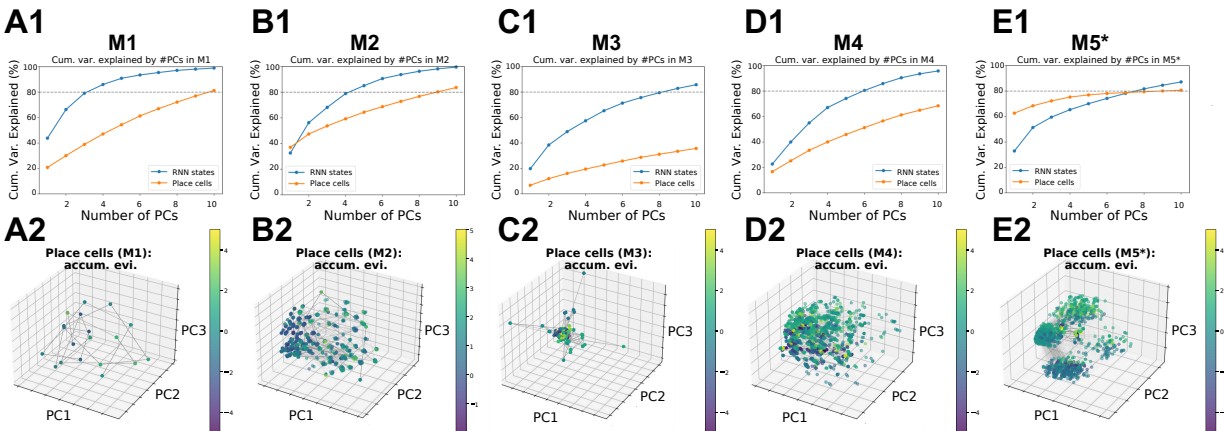

*Figure 11.* Cumulative variance explained in percentage of hippocampal (orange) and RNN (blue) activities, by the number of principle components (PCs), and low-dimensional representations of hippocampal activities in PC space, colored by accumulated evidence, shown for M1 (column A), M2 (column B), M3 (column C), M4 (column D), and M5 (column E). The first two PCs in M5 explained the most amount of variance in hippocampal representations (68%) in comparison to other model variants. We do not observe any visual separability of accumulated evidence in the PC space of the first three PCs, as shown in the second row.

## F. Effect of CA3 Recurrence in M2 & M4

In this section, we demonstrate in M2 and M4, as a proof of concept, that the inclusion of CA3 recurrent connectivity in the HPC layer does not affect the general conclusions presented in the main paper. Specifically, the inclusion of CA3 recurrence in M2 or M4 does not induce the experimentally observed place cell phenomena (Nieh et al., 2021), producing similar results as if recurrence was absent (see Figs 4, 5, 7, 8, and 9).

To model CA3 recurrence, we incorporate additional recurrent connections within the HPC layer, $\mathbf{W}_{hh}$, updated through hebbian-like associative learning using $\vec{h}_{\text{mix}}(t)$ and $\vec{h}_{\text{mix}}(t+1)$, analogous to the learning update for $\mathbf{W}_{hs}$ and $\mathbf{W}_{sh}$ (see Eqns. 4 and 5). The activity of mixed hippocampal cells is then described by:

$$\vec{h}_{\text{mix}}(t+1) = \text{ReLU}[\mathbf{W}_{hs} \cdot \vec{s}(t) + \mathbf{W}_{hg} \cdot \vec{g}(t+1) + \mathbf{W}_{hh} \cdot \vec{h}_{\text{mix}}(t)]. \tag{8}$$

The rest of the setup remains consistent with Appendix A.1. The above changes would be the same for models with $\vec{p}_{\text{nonmix}}(t)$ and $\vec{h}_{\text{nonmix}}(t+1)$.

As shown in Fig. 12, the inclusion of the recurrent integration of positional information from MEC and sensory information from non-grid EC does not result in the emergence of conjunctive place cells in M2. Similarly, Fig. 13 shows that the lack of conjunctive place cells in M4 persists when the grid cell modules encode position and evidence disjointly.

These findings confirm that the recurrent integration in HPC alone does not induce conjunctive coding, underscoring the critical role of joint integration of position and evidence in grid cells for producing co-tuned place cells.

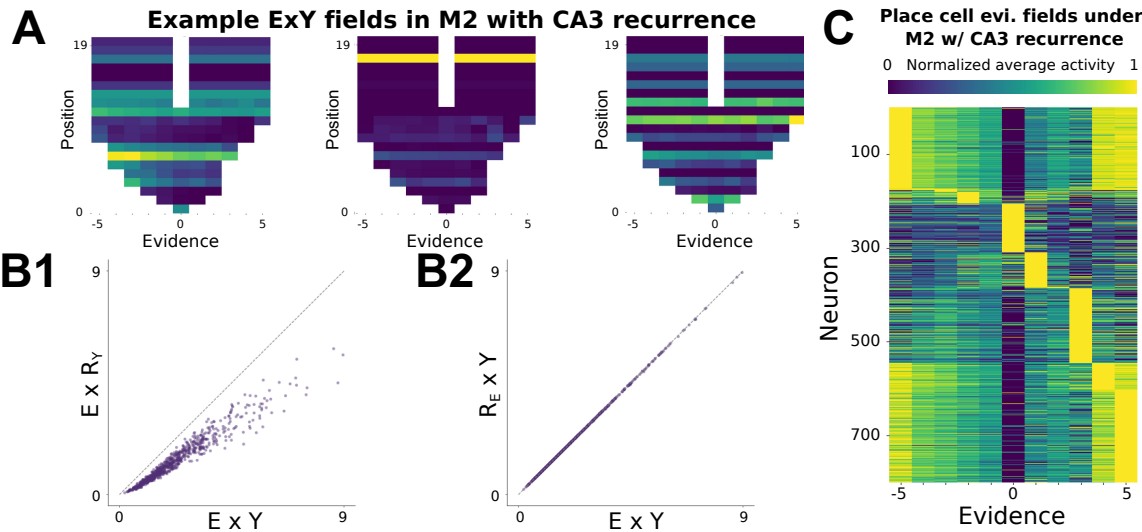

*Figure 12.* **Analysis of hippocampal code in M2 with CA3 recurrence.** **(A)** Example hippocampal firing fields in $E \times Y$ space. **(B)** Scatterplots of the hippocampal mutual information in M2 with CA3 recurrence when only the position is randomized (B1), and when only the evidence is randomized. The model shows higher mutual information in position only. See the caption of Fig B.2 for implementation details. **(C)** Hippocampal firing fields in evidence in M2 with CA3 recurrence.

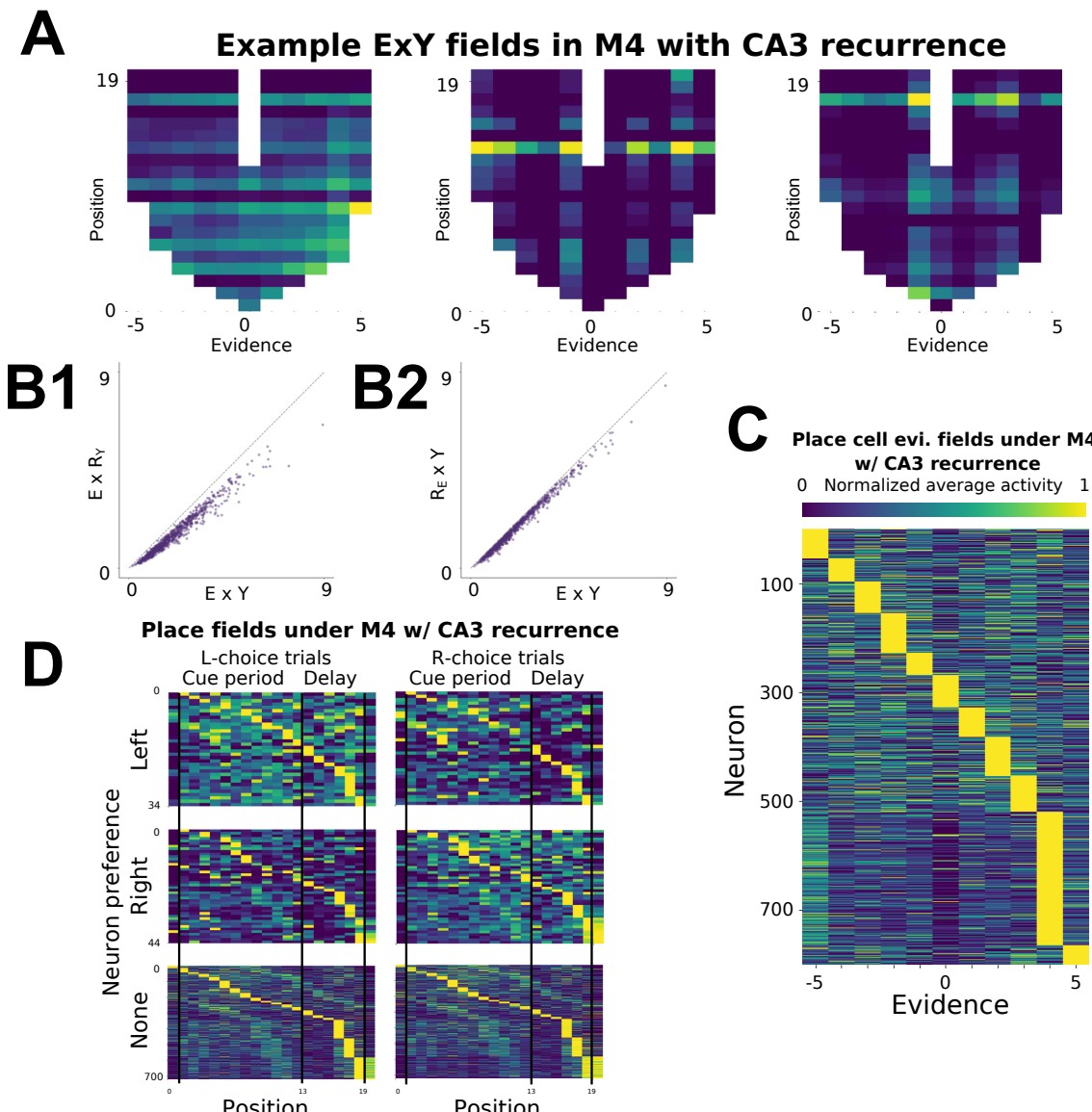

*Figure 13.* **Analysis of hippocampal code in M4 with CA3 recurrence.(A)** Example hippocampal firing fields in $E \times Y$ space. **(B)** Scatterplots of the hippocampal mutual information in M4 with CA3 recurrence when only the position is randomized (B1), and when only the evidence is randomized. The model shows higher mutual information in both position and evidence, consistent with the case when the recurrence is not considered (Fig B.2, column D). See the caption of Fig B.2 for implementation details. **(C)** Hippocampal firing fields in evidence in M4 with CA3 recurrence. We still observe evidence fields. **(D)** Hippocampal firing fields in space. We do not observe choice-specific place fields shown in Nieh et al. (2021) after considering the CA3 recurrence in M4.

## G. Learning Performance Under Hyperparameter Tuning

To evaluate the impact of hyperparameter choices on model learning, we conducted a sweep over learning rates (LR) for all models, including [5e−5, 1e−4, 5e−4, 1e−3], except when a specific learning rate was already selected in the main text, in which case we retained it without sweeping to ensure consistency with the main text. The performance curves with hyperparameter tuning results are visualized in Fig. 14.

Table 3 reports the mean final success rate and average exploration time (measured at the last 100 episodes across three independent trials), formatted as: `success[%] ± standard deviation / steps per episode ± standard deviation`. The best-performing metrics for each model are highlighted in bold.

*Table 3.* Mean success rate $\pm$ standard deviation and mean exploration time $\pm$ standard deviation across 3 trials over the last 100 out of 17400 episodes. Only the best success rate and lowest exploration time per model are in bold. The maximum number of steps allowed per episode is 200 steps. Our final model, M5, is fairly robust to learning rates as shown.

| LR | | M0 | M0+ | M1 | M2 | M3 | M4 | M5 |
|---|---|---|---|---|---|---|---|---|
| 5e−5 | Success (%) | $59.33_{\pm 11.73}$ | $72.00_{\pm 4.97}$ | $49.33_{\pm 2.87}$ | $50.33_{\pm 4.19}$ | $95.67_{\pm 2.05}$ | $93.67_{\pm 2.05}$ | $95.00_{\pm 1.63}$ |
| | Steps/ep. | $31.23_{\pm 2.46}$ | $27.94_{\pm 3.20}$ | $24.77_{\pm 0.66}$ | $\mathbf{21.33_{\pm 0.80}}$ | $28.91_{\pm 0.68}$ | $26.09_{\pm 1.67}$ | $25.71_{\pm 0.70}$ |
| 1e−4 | Success (%) | $\mathbf{68.67_{\pm 15.46}}$ | $\mathbf{81.67_{\pm 4.03}}$ | $47.33_{\pm 3.40}$ | $49.00_{\pm 3.56}$ | $\mathbf{97.00_{\pm 2.45}}$ | $\mathbf{99.33_{\pm 0.94}}$ | $\mathbf{99.00_{\pm 0.82}}$ |
| | Steps/ep. | $\mathbf{25.22_{\pm 1.76}}$ | $\mathbf{22.95_{\pm 1.18}}$ | $23.97_{\pm 0.81}$ | $23.01_{\pm 0.66}$ | $29.83_{\pm 1.45}$ | $\mathbf{25.52_{\pm 0.87}}$ | $27.65_{\pm 1.45}$ |
| 5e−4 | Success (%) | $8.00_{\pm 11.31}$ | $0.00_{\pm 0.00}$ | $51.33_{\pm 4.64}$ | $49.00_{\pm 2.45}$ | $96.33_{\pm 1.25}$ | $76.67_{\pm 16.54}$ | $\mathbf{97.00_{\pm 4.24}}$ |
| | Steps/ep. | $193.48_{\pm 9.22}$ | $200.00_{\pm 0.00}$ | $\mathbf{23.31_{\pm 0.38}}$ | $29.14_{\pm 8.35}$ | $\mathbf{23.34_{\pm 0.77}}$ | $28.02_{\pm 6.18}$ | $\mathbf{19.70_{\pm 0.22}}$ |
| 1e−3 | Success (%) | $0.00_{\pm 0.00}$ | $12.33_{\pm 17.44}$ | $\mathbf{51.67_{\pm 4.19}}$ | $\mathbf{51.67_{\pm 4.19}}$ | $67.33_{\pm 17.46}$ | $83.33_{\pm 19.48}$ | $90.00_{\pm 8.29}$ |
| | Steps/ep. | $200.00_{\pm 0.00}$ | $193.36_{\pm 9.39}$ | $25.81_{\pm 1.35}$ | $24.86_{\pm 2.13}$ | $33.71_{\pm 3.49}$ | $28.79_{\pm 4.02}$ | $30.74_{\pm 2.85}$ |

Across models, we observe that the chosen hyperparameters in the main text (1e−4 for M0 and M0; 5e−4 for M1-M5) generally yielded near-optimal performance. For M4, an LR of 1e−4 mitigated instability noted in the main paper's Fig. 3. This updated choice is applied to generate Fig. 14.

**Conclusion** Although learning rate tuning helped stabilize M4, none of these modifications altered the qualitative conclusions or key claims presented in the main paper. Section 5.1's observations regarding learning efficiency and rapid exploration in M5 remain unchanged.

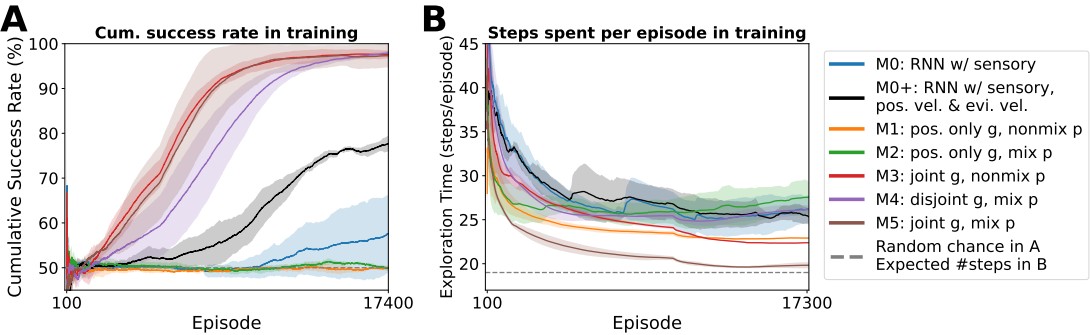

*Figure 14.* Updated Fig. 3 with optimized learning rate for each model, with a change to using a learning rate of 1e−4 for M4. The rest remains consistent with Fig. 3, and this does not alter the claims made in the paper. The moving average setup is the same as Fig. 3, including the window sizes used (5,000 for A and 10,000 for B) and the exclusion of the first 100 episodes.

## H. Results for Controlling Trainable Parameter Count in RNN Baselines

Here we provide additional ablations of the RNN baselines (M0 and M0+) to address whether the later-onset performance of M0 and M0+ might simply reflect a larger number of weights to optimize. We conclude that the performance of RNN baselines with the same number of parameters as M5, with or without additional velocity input, does not alter the conclusions drawn in the main paper.

**Parameter formula**    For a vanilla RNN with input dimension $I$, hidden dimension $H$, and output dimension $O$, the total number of *back-prop-trainable* parameters, including biases, is

$$\#\text{params} = \underbrace{(IH + HH + H)}_{\text{input \& recurrent weights + hidden bias}} + \underbrace{(HO + O)}_{\text{output weights + output bias}} = H^2 + H(I + O + 1) + O. \tag{9}$$

**Matching M0 or M0+ to M5**    Model M5 (with an RNN of input size $I = 800$, hidden size $H_{M5} = 32$, output size $O = 3$) contains

$$\#\text{params}_{M5} \approx 26\,755.$$

Solving Eq. (9) for $H$ with $I_{\text{mini-M0}} = 10$ and $I_{\text{mini-M0+}} = 12$, as well as the same $O = 3$ yields

$$H_{\text{mini-M0}} \approx 158, \qquad H_{\text{mini-M0+}} \approx 157,$$

so that the *mini* versions of M0 and M0+ have the same order of trainable parameters as M5.

**Control experiment**    We trained these mini-models for three independent trials (learning rate $10^{-5}$). Figure 15 shows their learning curves alongside the original M0/M0+ and M5. *The qualitative conclusion of the main paper is unchanged:* even when parameter counts are matched, M0 and M0+ still lag behind M5 in both learning speed and final performance.

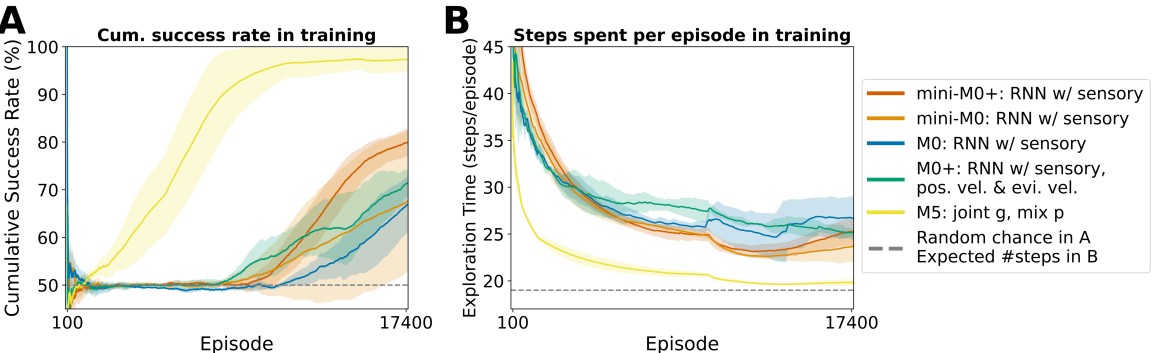

*Figure 15.* **Parameter count alone does not explain performance differences.** Learning curves for parameter-matched mini-M0 and mini-M0+ (red and orange), compared with the original M0 (blue), M0+ (black), and M5 (brown). **(A)** Cumulative success rate during training. **(B)** Steps spent per episode (exploration time). The plotting setup and color conventions match Fig. 3.

**Additional learning-rate sweeps**    We further trained the mini-models under alternative learning rates $\{1e{-}3, 5e{-}4, 5e{-}5\}$. The supplementary results (see Fig. 16) likewise show no qualitative change, reinforcing that parameter count alone does not explain the performance gap.

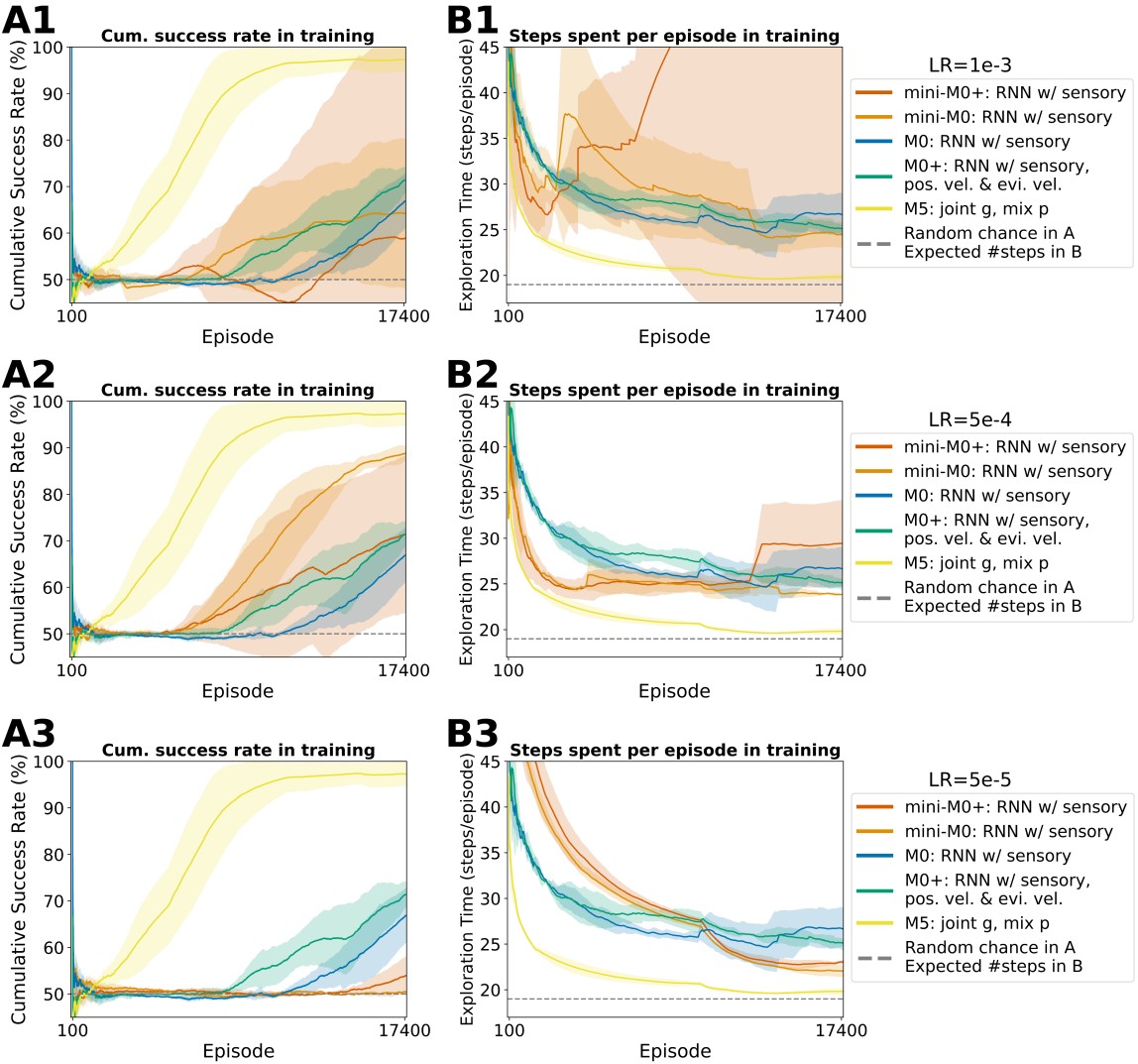

*Figure 16.* **Mini-models consistently underperform across learning rates.** Additional learning curves for parameter-matched mini-M0 and mini-M0+ trained with varying learning rates. **(A)** Cumulative success rate and **(B)** exploration time for mini-M0 and mini-M0+ (red and orange), which have approximately the same number of trainable parameters as M5 (brown). Original M0 and M0+ (blue and black) are shown for comparison. The plotting format and color scheme follow Fig. 3. Each row corresponds to a different learning rate used for the mini-models, indicated above the legend: $1e-3$ (top), $5e-4$ (middle), and $5e-5$ (bottom).

# I. Effect of smoothing on HPC tuning visualization

Here we provide additional clarification and visual evidence regarding the appearance of conjunctive tuning in the hippocampus neurons, particularly in models M4 and M5.

It was noted that the tuning curves shown in Fig. 4 appear weaker and more diffuse than those observed in Nieh et al. (2021). However, here we show that this difference can largely be attributed to differences in preprocessing and visualization–particularly the use of smoothing.

**Smoothing improves the appearance of conjunctive tuning**   Smoothing neural data is a fairly standard procedure empirically, which is done in both Nieh et al. (2021) and Chandra et al. (2025). When we apply smoothing procedures, the tuning curves become substantially more localized and stereotyped. Specifically, we follow the 2-stage filtering process described in Nieh et al. (2021):

1. Apply a 1D Gaussian filter with standard deviation $\sigma_1$.

2. Threshold the result by zeroing all values below a fixed multiple of the standard deviation across time.

3. Apply a second 1D Gaussian filter with $\sigma_2$.

We show examples of tuning curves after smoothing in selected HPC units from models M4 and M5 in Fig. 17. As anticipated, smoothing reveals structure that is otherwise obscured, making the tuning appear more consistent with findings reported in experimental work, such as Nieh et al. (2021).

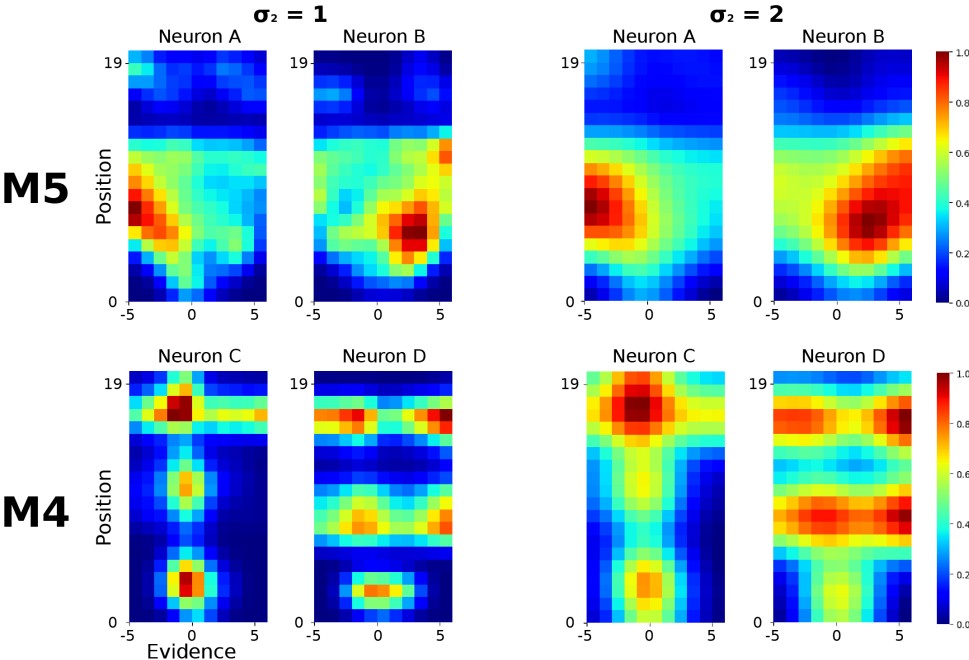

*Figure 17.* **Smoothing enhances visualization of conjunctive tuning.** Tuning curves from selected hippocampal neurons in M5 (top row) and M4 (bottom row) after applying a two-stage smoothing and thresholding procedure. Each row shows two example neurons (A, B in M5; C, D in M4), with each neuron visualized under $\sigma_1 = 1$, then two different levels of secondary smoothing: $\sigma_2 = 1$ (left two columns) and $\sigma_2 = 2$ (right two columns). Increased $\sigma_2$ leads to more diffuse but still conjunctive tuning. Smoothing reveals structure that is less apparent in raw activity and produces hippocampal fields more consistent with experimental observations.

**Relation to overall findings**   While smoothing improves the interpretability of individual neuron tuning curves, our broader conclusions regarding conjunctive representations are supported by quantitative measures such as mutual information (Appendix. B.2) and the presence of both place fields and evidence fields in HPC (Figs 5, 9). Thus, the visual appearance of raw tuning curves does not reflect a fundamental limitation of the model, but rather a visualization artifact that can be addressed through appropriate preprocessing.

