# OpenReview forum: "A Multi-Region Brain Model to Elucidate the Role of Hippocampus in Spatially Embedded Decision-Making"
_ICML.cc/2025/Conference — ICML 2025 poster_

### Official Review · Reviewer_zdy4 · 2025-03-06

**Overall Recommendation:** 4

**Summary:**

This work studies recent neurophysiological results through the use of computational modeling and reinforcement learning. The authors consider several different variants of the model (varying the connectivity and coding properties) and find only certain of these models achieve high performance and similarities to neural activations. The authors' work provides several direct predictions that can be tested with new experiments. More generally, the authors' approach proscribes a framework for studying decision-making embedded in spatial navigation.

**Claims And Evidence:**

All claims are supported by some evidence. However, I felt that M3 and M5's conjunctive coding of evidence and position (Figs. 3 and 7) was not convincingly shown. In particular, while it is clear that M1, M2, and M4 do no conjunctively encode evidence and position, the maps of M3 and M5 are not particularly localized (and differ from the maps of Nieh et al. (2021)). The authors compute the mutual information between E x Y and E x RY (and RE x Y) in Fig. 6 - could they do the same with the hippocampal data from Nieh et al. (2021) and compare the distributions? Something to make it more quantifiable how similar the ExY coding is in the model to the hippocampal data would strengthen this claim.

**Essential References Not Discussed:**

The authors include all "essential" references. However, I do think there are several relevant papers that are worth citing (which, I imagine the authors are familiar with):

1. https://journals.plos.org/ploscompbiol/article?id=10.1371/journal.pcbi.1007796 - The joint encoding of space and evidence reminds me of this work on how grid cells can encode higher dimensional variables. The random projection method discussed by Klukas and co-authors would be interesting to consider in the context of the authors model.

2. https://proceedings.neurips.cc/paper_files/paper/2017/file/5f14615696649541a025d3d0f8e0447f-Paper.pdf - The use of RNN models to generate hypotheses about navigation seems related to the work of  Kanitscheider and Fiete.

3. https://www.cell.com/neuron/fulltext/S0896-6273(11)00609-X (and other time cell related papers) - The idea that hippocampal neurons encode accumulating evidence (and temporal integration) is somewhat reminiscent of hippocampal cells encoding accumulating time.

**Experimental Designs Or Analyses:**

The experimental design and analysis were well done and appeared sound.

**Methods And Evaluation Criteria:**

The methods and evaluation make sense for the problem and were well motivated.

**Other Comments Or Suggestions:**

1. Both $p$ and $h$ are used to reference the hippocampal units (i.e., $W_{pg}$ and $W_{hg}$).
2. (Very minor) I had not heard of the word "lacunae" before. Maybe change it to something more common?
3. "theoretical work can leverage this standardized yet rich task and readily test its predictions" - should this be "and its predictions can be tested"?

**Other Strengths And Weaknesses:**

**Strengths:**
1. This paper provides a general framework for testing hypothesis on hippocampal-MEC-cortical interactions.
2. This paper is able to replicate existing neurophysioloigcal results and leads to new predictions on the role of grid cells that can be experimentally tested.
3. This paper is generally well written and motivated, and the experiments are well done.

**Weaknesses:**
1. As noted previously, the major weakness to my mind is that the conjunctive encoding of location and evidence is not convincing from the heat maps alone. Quantifying the extent of this joint encoding would strengthen this claim.
2. The results in Fig. 2A clearly show that only M3 and M5 are able to optimally perform the task. This has clear implications for building an AI system that can solve the task. But the mice in Nieh et al. (2021) don't perfectly solve the task. Looking at Fig. 1b in Nieh et al. (2021), it seems like the mice are solving the task at around 75% accuracy. M0, M0-star, and M4 all solve the task at around 70% at around by the end of training. I understand that the subsequent results provide additional support for M3/M5 having similarities to the neural recordings, but discussing this aspect of performance (that the mice did not perform the task as well as M3/M5) is important I think.
3. Models M0 and M0-star have the same number of units as the other models. But one thing that was not clear to me was if they have the same number of trainable weights (since the Vector-Hash model has some fixed weights). Could one reason models M0 and M0-star not perform as well until later in training be due to the fact that they have more weights to optimize over?

**Questions For Authors:**

1. How does the mutual information distributions (for E x Y) compare between real hippocampal data and the models?
2. How closely do models M0, M0-star, and M4 perform to the mice in Nieh et al. (2021)?
3. How does the number of training able parameters differ between M0/M0-star and the other models?

**Relation To Broader Scientific Literature:**

The paper explores a surprising neurophysiological result that challenges how the field thinks about place cells and their role in spatial navigation and episodic memory. The authors' result, finding that they expect grid cells to jointly encode space and evidence, is itself surprising and motivates greater study of the MEC-hippocampus-cortex circuit. These points were well made by the paper and the Introduction did a nice job of laying out the motivation.

**Theoretical Claims:**

No theoretical claims were made.

---

> ### Author Rebuttal · Authors · 2025-03-29
>
> We sincerely thank the reviewer for their precious time and comprehensive feedback. We deeply appreciate their recognition of the significance of our work. In summary, the reviewer raised an insightful question regarding experimental v.s. model data comparison to strengthen our claims, and our interpretation of task performance, for which we discuss the implication of M5 for studying lapses. We provide results for when we shrink the number of trainable parameters of baselines to match that of M5, which do not alter our conclusion. We have fixed the typos on our end. **Together with the discussion below, does this address the reviewer’s questions?**
>
> > The authors compute the mutual information…in Fig. 6. could they do the same with the hippocampal data from Nieh et al. (2021) and compare the distributions? Something to make it more quantifiable…would strengthen this claim.
>
> Thanks for the insight. We’d like to first refer to Figs S2b, c in Nieh et al., which plot mutual information (M.I.) of hippocampal data, the same as our Fig 6. We share it here for convenience: https://imgur.com/a/acEavag.
>
> This qualitatively matches the results of M3-M5 in Fig 6, showing models with both position and evidence encoded in grid cells give rise to higher ExY M.I. than when one variable is randomized. This makes sense for M4 because M.I. is not a metric for localization. However, as shown in Fig 4, M4 does not match experiments in having choice-specific neurons.
>
> We don’t expect a quantitative match in M.I, because
> 1. Our environment is discretized;
> 2. We don’t have the same level of noisiness as the experiments;
> 3. Real neurons exhibit higher redundancy;
> 4. The real data is smoothed but we didn’t apply smoothing, hence the scale of M.I. is different. The smoothing also influences the localization in HPC maps.
>
> > M0, M0-star, and M4 all solve the task at around 70% at around by the end of training…subsequent results provide additional support for M3/M5…but discussing this aspect of performance (that the mice did not perform the task as well as M3/M5) is important I think.
>
> Indeed, the lapses phenomenon in mice (low performance) is an important ongoing research direction.
>
> While early literature proposed lapses could be due to perceptual noise, more recent work like Ashwood et al., Nat Neurosci., 2022 pointed out that mice might be leveraging different strategies characterized by different states, while Pisupati et al., eLife, 2021 proposed the mice are potentially balancing exploration and exploitation when making mistakes. These statistical models have specific parameters to account for lapses based on their hypotheses and are evaluated based on how well they fit the data. While we find the lapses literature important, we don’t think the low performance of M0, M0+, & M4 would be informative for understanding lapses, as you carefully noted that they are limited in reproducing experimental findings.
>
> While studying lapses is beyond our scope, future studies on lapses could leverage M5, validated as being able to reproduce experimental findings. It’d be interesting to investigate what the key ingredients are to observe lapses while preserving the properties we observed. To our knowledge, a mechanistic model for lapses would be fairly novel. **We will include this in our discussion.**
>
> > Could one reason models M0 and M0-star not perform as well until later in training be due to the fact that they have more weights to optimize over? How does the number of training able parameters differ between M0/M0-star and the other models?
>
> This is a fair point. We focused on an equal number of neurons as a fair baseline for expressivity. Here we found the number of parameters trained by back prop doesn't play a key role:
>
> Assume the total number of parameters in an RNN w/ input size $I$, hidden size $H$, and output size $O$, w/ bias, is $(IH + HH + H) + (HO + O) = H^2 + H(I+O+1) + O,$ then M5 has ~$26,755$ back-prop-trainable parameters. We can apply similar reasoning to M0 and M0+, with $1,172,843$ and $1,174,995$ gradient-trainable parameters. To match M5, M0 would have $H \approx 158$ and M0+ would have $H \approx 157$.
>
> We ran these mini-models for 3 trials each: the **conclusion in the main paper stays the same**. Plz see the anonymous results here with mini-M0 & mini-M0+ (red and orange, w/ learning rate of 1e-5), in comparison with M5 and original M0 and M0+: https://imgur.com/a/Qr5MmT4
>
> Supplement: mini models w/ other learning rates, no change of conclusions: https://imgur.com/a/vCD2HRG
>
> We notice our use of M0-star & M0+ interchangeably; we will use “M0+” consistently.
>
> > Other Comments
>
> Thanks! We'll ensure a consistent notation `h`. We’ll modify “lacunae” to gaps and correct the sentence.
>
> ---
> Ashwood, Zoe C., et al. "Mice alternate between discrete strategies during perceptual decision-making." Nature Neuroscience (2022).
>
> Pisupati, Sashank, et al. "Lapses in perceptual decisions reflect exploration." Elife (2021).

---

> > ### Comment · Reviewer_zdy4 · 2025-04-04
> >
> > I thank the authors for their detailed responses to my questions. I apologize for not noting the MI plot in Nieh et al. - thank you for pointing me to it. I am convinced by the additional experiments with the change in number of trainable parameters for M0 and M0+. And I am glad to hear the authors will add more discussion on lapse of performance - I agree this is a really interesting future direction!
> >
> > My one remaining question is (which I should have put in the question section) is why do the authors think the conjunctive tuning in the RNN is not as strong as in Nieh et al. Indeed, to me it looks like the tuning in Fig. 3 is quite weak and diffuse. If you applied smoothing, would they more like what was seen in Nieh et al.? Or maybe some aspect of the task needs to be changed for stronger tuning? Any thoughts on this and any added discussion in the paper on the fact that the tuning is weak would be appreciated.

---

> > > ### Author Response · Authors · 2025-04-07
> > >
> > > We thank the reviewer for their prompt and insightful follow-up. We are glad that the reviewer’s questions regarding lapses and the number of trainable parameters in baselines are resolved.
> > >
> > > Here we address the follow-up regarding the tuning visualization of HPC activities:
> > > > My one remaining question is (which I should have put in the question section) is why do the authors think the conjunctive tuning in the *HPC* is not as strong as in Nieh et al. Indeed, to me it looks like the tuning in Fig. 3 is quite weak and diffuse. If you applied smoothing, would they more like what was seen in Nieh et al.?
> > >
> > > In short, **yes, if we apply smoothing, the conjunctive tuning in HPC looks more localized and stereotypical.** Notably, smoothing neural data is fairly standard in literature, likely for visualization purposes. For example, the Vector-HaSH paper smoothed and interpolated the HPC activities when showing the tuning curve for better visualization (see Fig 4b in Chandra et al., 2025 and their official [code repository](https://github.com/FieteLab/VectorHaSH/blob/main/Grid_place_tuning_curves_and_additional_expts_Fig1_4_6.ipynb)).
> > >
> > > Here we demonstrate that applying smoothing enhances the localization of tuning curves in selected neurons from M4, M5: https://imgur.com/a/hIeKSar.
> > >
> > > We follow a similar 2-stage processing procedure in Nieh et al. (Mutual Info Analysis in Method): we apply 1d Gaussian filter with a $\sigma_1$ of $1$, then thresholded the result so that values less than $2$ standard deviations across the time series were set to $0$; we then apply 1d Gaussian filter with a $\sigma_2$ of $1$ or $2$. We will add the above figure to our Appendix. We appreciate the reviewer’s comment.
> > >
> > > We’d like to emphasize that our mutual information analyses (Fig 6) and the place fields and evidence fields (Figs 4 & 8) provide supporting evidence on the overall HPC (conjunctive) tuning of each model, despite smoothing impacts the quality of activity visualization.

---

### Official Review · Reviewer_QCkJ · 2025-03-06

**Overall Recommendation:** 4

**Summary:**

This paper introduces a series of models to investigate how animal brains may solve the accumulating towers task. Beginning with a simple RNN, the authors add model components until they arrive at extensions of the Vector-HaSH model of the hippocampus that also include a cortical model component. The authors now evaluate the models on their ability to solve the accumulating towers task, but also analyze how the added complexity enable to increased model performance.

The authors predict that only models that include models that allow conjunctive position-evidence tuning in grid cells exhibit the conjunctive position-evidence hippocampal representations that had been identified earlier experiments in mice.

## update after rebuttal

I have no remaining questions for the authors. I applaud that the authors commit to releasing the code for their experiments, and are putting effort into making their code understandable and useful for other researchers.

**Claims And Evidence:**

Yes, the main claims are clearly laid out and supported well.

**Essential References Not Discussed:**

Relevant literature has been cited.

**Experimental Designs Or Analyses:**

The experiments seem sound.

It would be helpful if the authors could include more information on how hyperparameters for model training were chosen, and how much those hyperparameters affect each of the half-dozen models that the authors built. A few extra sentences in appendix A.1 should be sufficient.

It would be helpful if the authors could include more equations in the paper, likely in an appendix. For example the CAN equations should be included.

**Methods And Evaluation Criteria:**

Yes, the accumulating towers task is suited well for this investigation.

**Other Comments Or Suggestions:**

-  It would be helpful if the authors could include more equations in their paper. For example, the CAN or Vector-HaSH equations (which have of course been published elsewhere in papers cited by the paper reviewed here) are so central to the paper that they should in my opinion by included in the paper or in the appendix.

**Other Strengths And Weaknesses:**

Strengths:
- The paper advances the field of hippocampal modeling by investigating how state-of-the-art existing models of the hippocampus might perform when integrated with other parts of the brain. While the work reviewed here is entirely theoretical (computer models), the work appears to be inspired by recent (2021) measurements taken in brains of mice. The paper’s alignment with existing laboratory measurements on the task that is being investigated (accumulating towers) makes this work particularly exciting. Last: the authors announce in the paper that they are performing experimental work to test hypotheses born out of the models built here.
- The work presented in the paper is quite comprehensive: the authors investigate half a dozen models, all of which are interesting from a computational neuroscience perspective.
- The paper is written well: while the paper is dense, it does a good job of conveying the material. The diagrams showing the model architectures make it easy to understand the experimental set-up.
- The authors highlighted that their work builds on the Vector-HaSH method published earlier. This enables readers who are familiar with the surrounding literature to instantly situate the paper.

Weaknesses:
- It would be helpful if the authors could include more equations in their paper. For example, the CAN or Vector-HaSH equations (which have of course been published elsewhere in papers cited by the paper reviewed here) are so central to the paper that they should in my opinion by included in the paper or in the appendix.
- While the authors promise to release the code for the experiments, they did not include them with the supplementary materials. While authors may be worried about their code getting stolen by reviewers, peer review can inspire improvements to experiment code that benefit the quality of the paper, and make it easier for other research groups to build on the paper, or even just benchmark against the models presented here.

**Questions For Authors:**

The authors mention on line 682 (Appendix A.1) that different learning rates were used for different model configurations, and that a hyper parameter search was performed for some configurations, while other configurations used common “default” hyper parameter configurations. This made me wonder: how much would tuned hyperparameters have helped for models that used default hyper parameter configurations?

**Relation To Broader Scientific Literature:**

This paper uses the Vector HASH method https://www.biorxiv.org/content/10.1101/2023.11.28.568960v2 , a recent and state-of-the-art hippocampal model, as a core component in most of the models investigated. The authors build a series of models and compare their behavior with behavior that was observed in laboratory experiments on mice in earlier work. These models present significant extensions of the Vector HASH method, and advance our understanding of the hippocampus and brain regions that it connects to.

The paper is particularly exciting because the authors pay close attention to existing neuroscience literature, and marry this with complex but meaningful modeling.

**Theoretical Claims:**

There are no theorems or proofs.

---

> ### Author Rebuttal · Authors · 2025-03-29
>
> We sincerely thank the reviewer for their precious time, and their comprehensive feedback. We deeply appreciate their recognition of the significance and clarity of our work. In summary, the reviewer raised an insight regarding the role of hyperparameters, and made additional comments to help us enhance clarity. We share the hyperparameter tuning results and CAN implementation, which are added to the appendix. **Together with the discussion below, does this address the reviewer’s questions?**
>
> > It would be helpful…how hyperparameters for model training were chosen…A few extra sentences in appendix A.1 should be sufficient.
> > how much would tuned hyperparameters have helped for models that used default hyper parameter configurations?
>
> Thank you for attention to this detail. M0 and M0+ used a different learning rate due to gradient issues (line 681). We share the result on hyperparameter tuning on learning rate (LR) [0.001, 5e-4, 1e-4, 5e-5] (here’s an anonymous colored version: https://imgur.com/a/Ve7B3Mo).
>
> The current set of hyperparameters largely ensures fairly optimized performance, except a slower LR of 1e-4 improves M4 instability in Fig 2. **However, none of them changed the claims made in the paper.** The table includes mean [success rate]/[exploration time] +- sem at the terminating episode across 3 trials. The highest success rate and the lowest exploration time are bolded for each model.
>
> | LR      | M0                             | M0+                            | M1                              | M2                              | M3                              | M4                              | M5                              |
> |--------:|:------------------------------:|:------------------------------:|:-------------------------------:|:-------------------------------:|:-------------------------------:|:-------------------------------:|:-------------------------------:|
> |0.00005  |61.69±9.27/27.49±3.05           |**74.13±5.90**/**24.80±0.22**   |49.98±0.24/24.63±0.68            |**50.34±0.78**/**21.80±0.47**    |93.76±1.28/28.88±0.42            |92.73±3.69/**25.23±0.93**        |95.03±0.50/25.94±0.79            |
> |0.0001   |**67.96±5.65**/**26.65±2.41**   |72.54±1.40/25.13±0.60           |49.75±0.20/25.39±0.56            |50.19±0.21/23.33±0.96            |94.59±2.50/29.11±0.17            |**97.35±0.34**/26.53±0.70         |**97.78±0.55**/27.04±0.67         |
> |0.0005   |0.00±0.00/200.00±0.00           |0.00±0.00/199.99±0.02           |49.99±0.55/**22.92±0.04**        |49.79±0.46/27.66±1.77            |**97.62±0.87**/**22.42±0.04**    |72.13±17.06/25.47±4.11           |97.52±2.40/**19.96±0.22**        |
> |0.001    |0.00±0.00/200.00±0.00           |0.00±0.00/200.00±0.00           |**50.18±0.21**/24.64±0.34        |49.95±0.29/26.67±1.84            |83.86±8.77/32.60±1.13            |55.11±7.09/28.14±1.84            |87.23±12.14/27.48±1.29           |
>
> We also provide **an updated Fig 2** with the optimized LRs. The changes are a LR of 5e-5 for M0+ and 1e-4 for M4. This **doesn’t alter the conclusion** in Section 5.1 regarding M5’s learning efficiency & fast exploration: https://imgur.com/a/U0OVo2H. **We'll include this in the appendix.**
>
> > It would be helpful if the authors could include more equations in the paper, likely in an appendix. For example, the CAN equations...
>
> Thank you for pointing this out. We agree that including more equations would enhance the clarity. If accepted, we will include below in the appendix in the camera-ready version. Specifically, we implement $CAN()$ in Eq. (1), based on the Vector-HaSH repo [1]:
>
> $g(t+1) = \textsf{CAN}[g(t), v(t)]
>  = \mathbf{M} g(t),$
>
> where $\mathbf{M}$ denotes a shift matrix depending on the velocity signal $v$. For simplicity, suppose we have two grid cells with periodicities 3 and 4 and use a single dimension instead of two dimensions (our case). Then, $\mathbf{M}$ is defined as a shift matrix in each grid module as follows:
>
> $
> M = U = \begin{bmatrix}
> 0 & 1 & 0 & 0 & 0 & 0 & 0 \\\\
> 0 & 0 & 1 & 0 & 0 & 0 & 0 \\\\
> 1 & 0 & 0 & 0 & 0 & 0 & 0 \\\\
> 0 & 0 & 0 & 0 & 1 & 0 & 0 \\\\
> 0 & 0 & 0 & 0 & 0 & 1 & 0 \\\\
> 0 & 0 & 0 & 0 & 0 & 0 & 1 \\\\
> 0 & 0 & 0 & 1 & 0 & 0 & 0 \\\\
> \end{bmatrix}
> $
>
> if $v$ shifts the bump activity to the right, otherwise $U^T$.
>
> Formally, $M_{i,j} = 1$ if $(j - X_{k-1}) \mod \lambda_k \equiv (i + \text{velocity} - X_{k-1}) \mod \lambda_k$, where  $X_{k-1} < i, j < X_k  \forall i, j, k, X_k = \sum_{l=1}^{k} \lambda_l$. Otherwise, $M_{ij} = 0$.
>
> 2D Vector-HaSH is a simple extension of this to two-dimensional grid states and velocities.
>
> > code release.
>
> We appreciate the reviewer’s emphasis on reproducibility and agree accessible code is important for advancing the field. We're finalizing documentation and ensuring the code is well-organized and user-friendly. We remain committed to releasing the full codebase upon acceptance (w/ the camera-ready version).
>
> ---
>
> [1] https://github.com/FieteLab/VectorHaSH

---

### Official Review · Reviewer_gA4b · 2025-03-12

**Overall Recommendation:** 4

**Summary:**

This work is motivated to implement efficient reinforcement learning (RL) inspired by computation in hippocampus. It develops a multi-region brain model that incorporates hippocampal-entorhinal circuit based on the Vector-HaSH model. It shows a structured, content-addressable associative memory with neural representations is biologically grounded efficient RL solver. It proposes with a variants of models from M0 to M5 with different grid cell and place cell coding strategy, and inputs to the MLPs or RNNs. The results demonstrate joint integration model induces efficient learning, and evidence-position co-tuning in grid cells, how this model's results aligned with current experimental evidence.

**Claims And Evidence:**

This work is based on existing Vector-Hash model, and extending its studies to a multi-region brain model across entorhinal, hippocampal and neocortical regions, and systematically exploring the role of joint integration and coding strategy to induce efficient learning. It proposes a variant of models to find the one leads to the most efficient learning (number of episodes taken to converge). It also further checks how the finding from models aligned with existing experimental findings, which presents a very solid study.

**Essential References Not Discussed:**

N/A

**Experimental Designs Or Analyses:**

The soundness/validity of experimental designs have been evaluated, systematic explorations are conducted, and results are solid. One major limitation is that the work is mostly biologically driven, as it claims to an efficient brain-inspired reinforcement learning rule, while it does not directly compare with existing models in artificial intelligence sides.

**Methods And Evaluation Criteria:**

The study is conducted with a task-driven perspective, and evaluates with multiple variants of constraints, to finding alignments between experimental phenomenon, which is orthogonal to the data-driven paradigm, and directly fitting the experimental measured the data, which could also ignore critical components. And only one task has been evaluated in the experiment, this framework could potentially extend to more diverse tasks and settings to achieve a more comprehensive study.

**Other Comments Or Suggestions:**

N/A

**Other Strengths And Weaknesses:**

As described above, this work introduces a systematic exploration of the computation in hippocampus for decision making with a task-driven perspective, and it delivers interesting results on how a joint integration strategy contributes to efficient learning and aligning with existing experimental discovered phenomenons.

**Questions For Authors:**

Any model designs do not directly follow biological constraints, or findings are not aligned with experimental discoveries?

**Relation To Broader Scientific Literature:**

This work has covered sufficient amount of literatures and a comprehensive survey. The study is well grounded.

**Theoretical Claims:**

There are no theoretical proofs or claims.

---

> ### Author Rebuttal · Authors · 2025-03-29
>
> We sincerely thank the reviewer’s precious time and comprehensive feedback. We deeply appreciate their generous recognition of the significance of our work. The reviewer shared an interesting insight regarding comparison with advanced AI models, and had a clarifying question on how our setup and findings compare with the biology. Here we address both. **Together with the discussion below, does this address the reviewer’s questions?**
>
> > One major limitation is that the work is mostly biologically driven, as it claims to an efficient brain-inspired reinforcement learning rule, while it does not directly compare with existing models in artificial intelligence sides.
>
> Thanks a lot for the insights, and we agree having some comparison with advanced AI models could be quite interesting to see if brain-inspired models could perform SoTA in the standard machine learning benchmark (e.g, image classification, Atari). However, our main focus is to design a biologically-inspired model that can be applied to explain neuroscience experiments. This objective is aligned with previous neuroscience-application studies that only use simple RNNs as baselines published in top ML venues (e.g., Miller et al., NeurIPS, 2023; Valente et al., NeurIPS, 2022).
>
> > Any model designs do not directly follow biological constraints, or findings are not aligned with experimental discoveries?
>
> We think the biological constraints can be assessed through
> 1. the information flow among regions, and
> 2. the representation produced by the model.
>
> While there are not sufficient neuroscience experiments (to the best of our knowledge) revealing the exact information flow among regions in the context of spatially-embedded decision-making tasks, M1-M5 are all reasonable hypotheses of (1) given the current neuroscientific understanding (elaborated in Section 4). However, as presented in our paper, M1, M2, M4 did not produce experimentally aligned representations (e.g., Figs 4, 8).
>
> ---
>
> Miller et al. "Cognitive model discovery via disentangled RNNs." NeurIPS. 2023.
>
> Valente et al. "Extracting computational mechanisms from neural data using low-rank RNNs." NeurIPS. 2022.

---

### Official Review · Reviewer_c3tv · 2025-03-13

**Overall Recommendation:** 2

**Summary:**

This paper aims at providing a mechanistic characterisation of how sensory and abstract (task-dependent) information is encoded and transmitted across different brain regions, including the hippocampus (HPC), medial and lateral entorhinal cortex (mEC and lEC), and the cortical circuity. The authors specifically focus on explaining the data reported in Nieh et al., 2021, which is an decision-making task based on sensory evidence accumulation under the spatial context. The model is largely based on the existing Vector-HASH+ model (Chandra et al., 2025), but with the additional cortex component (represented by an RNN) that takes in the hippocampus readout and map into actions. Several variants of the baseline Vector-HASH model are proposed, underlying different candidate intra- and inter-circuitry mechanisms. The authors then trained the models on the abstracted version of the towering task from Nieh et al., 2021, and show that post-training, the place cells indeed exhibit selectivity with respect to spatial location and sensory evidence. They also show that the sensory and task information are only well-separated in one specific circuitry configuration, and making predictions for the mEC-HPC circuitry.

**Claims And Evidence:**

The paper is quite well-written, with comprehensive citations to relevant literature.

Some claims made by the authors are ungrounded and questionable, I list them and some general comments/questions below.

- The authors assumes direct projections from lEC and mEC consistutes the firing of place cells. How could the authors enforce sparsity and spatial selectivity in place cells then? I do not see a mechanistic explanation for the necessary occurrence of place fields. I think the missing of inductive biases in constructing the place cells limits the capability of the model for predictions beyond those already presented in Nieh et al. 2021.
- The fact that M0 and  M0+ models do not yield good performance could due to poor implementation of the recurrent structure. I can imagine a recurrent networks designed for temporal integration (e.g., those used for explaining drift diffusion) will be able to perform the task whilst not requiring the complicated EC-HPC network. I hence think this is an unfair comparison and statement to make.
- The authors make claims about the criticality of sensory inputs from lEC in driving efficient learning and spatial navigation. However, I cannot draw such conclusion from Figure 5. The lack of qualitative comparison with M3, the model that yields similar performance in the decision making task as the M5 model, undermines the validility of the authors' claim.
- The model is largely an adoption of the Vector-HASH+ under a simple RL setting. I do find the neural predictions and hypotheses interesting, but the paper is limited in terms of methodological novelty.
- It would be useful to apply the model to other tasks to substantiate the validity of the model, such as the two-armed bandit task from Mishchanchuk et al., 2024.

Overall, I accredit the authors' mechanistic attempt, but I personally find many claims made in the paper to be ungrounded, and no sufficient ablations exist to support their claims. Adding on the limited methodological novelty in the paper, I am leaning towards rejection. But I am happy to change my mind should the authors provide compelling empirical support for their claims.

**Essential References Not Discussed:**

No.

**Experimental Designs Or Analyses:**

Yes.

**Methods And Evaluation Criteria:**

Yes.

**Other Comments Or Suggestions:**

N/A

**Other Strengths And Weaknesses:**

N/A

**Questions For Authors:**

N/A, see above.

**Relation To Broader Scientific Literature:**

The paper proposes a mechanistic model for predicting circuitry configurations across multiple brain regions that lead to cognitive behaviours. I think this is a useful application of the Vector-HASH+ model beyond spatial contexts.

**Theoretical Claims:**

There is not theoretical claims in the paper.

---

> ### Author Rebuttal · Authors · 2025-03-29
>
> We deeply appreciate the reviewer’s precious time & comprehensive feedback. We sincerely thank them for recognizing the significance & clarity of our work. The reviewer raised questions on the support for our claims, method novelty, and the biological grounding of models. Here we clarify with the relevant content in the paper. **Together with the discussion below, does this address the reviewer’s concerns?**
>
> > The model is largely an adoption of the Vector-HASH+... I do find the neural predictions and hypotheses interesting, but…limited in terms of methodological novelty.
>
> Thanks for acknowledging our findings. The architecture is based on Vector-HaSH (VH, Chandra et al., Nature, 2025), but **the “+” part is our original development (lines 215-218)**. We emphasize our application-driven work is among the first to apply VH systematically, with appropriate adaptation (e.g., the “+” part), to understand spatial decision-making w/ experimentally verifiable predictions. Similar studies exist w/ RNNs but lack the necessary bio details in modeling multi-region circuits.
>
> > ...I accredit the authors' mechanistic attempt, but I personally find many claims made in the paper to be ungrounded, and no sufficient ablations exist to support...
>
> Our M1-M5 are ablations of multiple hypothesized multi-region interactions. We refer to VH (Chandra et al., 2025) for the detailed bio grounding as they’ve substantially addressed them. We ensured further claims are supported w/ evidence. We address below the reviewer's specific concern on the sensory criticality claim, but happy to elaborate on other claims if needed:
>
> > The authors make claims about the criticality of sensory inputs from lEC in driving efficient learning and spatial navigation. However, I cannot draw such conclusion from Fig 5. The lack of qualitative comparison with M3…undermines the validility of the authors' claim.
>
> The sensory criticality claim is made for efficient navigation & low-d representation, **not** for learning. The navigation aspect is evident in Fig 2B (M3 vs M5) & lines 304-314; low-d representation aspect is in Fig 5, w/ M3 in Appendix E, referred to in line 413. Joint grid code is critical to efficient learning (M3 & M5 in Fig 2A, lines 299-303).
>
> > The authors assumes direct projections from lEC and mEC consistutes the firing of place cells. How could the authors enforce sparsity and spatial selectivity in place cells?
>
> Relevant bio groundings are inherited from & addressed in VH (Fig 6 in Chandra et al., 2025), so we didn’t reiterate. We'll include these details in the appendix:
>
> **Sparsity:** Per line 200, $W_{hg}$ is a random projection matrix generated by *standard Gaussian distribution* (Method in Chandra et al. (2025)), so half of the activation is 0 on expectation. The input is sparse as the firing of each grid module is one-hot per model inductive bias (line 208).  ReLU is applied to `h` (eqns 2 & 3) to also enforce sparsity.
>
> The number of unique grid states ($\prod \lambda^2_i$) is much smaller than the number of unique activated HPC states ($2^{N_h}$), so only a small number of HPC cells are >0.
>
> **Selectivity:** Each sensory state is associated w/ a specific grid state by updating $W_{hs}, W_{sh}$, so each sensory state is only associated with certain HPC states.
>
> > …M0 and M0+ models do not yield good performance could due to poor implementation of the recurrent structure. I can imagine a recurrent networks designed for temporal integration…will be able to perform the task whilst not requiring the complicated EC-HPC network...think this is an unfair comparison...
>
> Vanilla RNN is fairly powerful & extensively studied, e.g., Yang et al., Nat Neurosci, 2019 & Driscoll et al., Nat Neurosci, 2024 applied RNNs to NeuroGym tasks with temporal components.
>
> And, our goal is to assess the added value of the structured EC-HPC networks, not to benchmark arbitrary network classes; changing the recurrent structure of baselines necessitates changing M1-M5’s RNN for fair comparison. This would be unnecessary and defeat the purpose of isolating the inductive bias introduced by EC-HPC circuits.
>
> Our work highlights the utility of leveraging VH as a **hypothesis-generating testbed** to study the entorhinal-hippocampal-cortical network. A task-optimized recurrent structure misses the necessary bio details.
>
>  > ...useful to apply the model to other tasks…validity of the model...two-armed bandit...
>
> The validity of the model is grounded in Chandra et al. (2025), and we emphasize the depth, not breadth, of results since
> 1. Our scope is spatially embedded decision-making with phenomena specific to HPC & related circuits. e.g., two-armed bandit task doesn't study the **spatial aspect** of HPC.
> 2. The tower task is justified for many reasons noted in lines 36-63, e.g., `we focus on this task to integrate our findings into a larger cohesive narrative that transcends the inherent scope limitations of stand-alone studies on arbitrarily chosen tasks`.

---

### Decision · Program_Chairs · 2025-05-01

**Decision:**

Accept (poster)

**Comment:**

This paper proposes a biologically grounded multi-region model of the hippocampus and entorhinal cortex integrated with cortical action selection to explain spatially embedded decision-making. It systematically compares circuit architectures using variants of the Vector-HaSH model and generates testable predictions about conjunctive grid cell coding of position and evidence. The authors apply their models to the accumulating towers task and show that a specific architecture, M5, reproduces neural and behavioral patterns observed in experimental studies.

The strengths of the paper lie in its neuroscientific motivation, strong alignment with experimental data, and comprehensive exploration of alternative circuit mechanisms. The authors provide ablations across multiple structured models, quantify neural representations, and respond thoroughly to reviewer concerns—especially regarding mutual information comparisons, parameter counts, and visualization quality. The commitment to code release and detailed clarifications in the rebuttal further demonstrate the authors’ thoughtfulness and rigor.

One reviewer expressed concerns over limited methodological novelty and interpretability of heatmaps, but these are addressed by explaining the adaptation of Vector-HaSH and adding smoothing analyses in the appendix. While the model does not outperform baseline RNNs on complex AI benchmarks, its purpose is hypothesis-driven modeling of biologically plausible decision-making circuits.

Given the paper's careful design, robust empirical evidence, biological relevance, and constructive engagement with reviewer feedback, I recommend acceptance.